# L-SCRaMbLE as a tool for light-controlled Cre-mediated recombination in yeast

Lena Hochrein[1,5], Leslie A. Mitchell[2], Karina Schulz[1,6], Katrin Messerschmidt[1] & Bernd Mueller-Roeber[3,4]

The synthetic yeast genome constructed by the International Synthetic Yeast Sc2.0 consortium adds thousands of loxPsym recombination sites to all 16 redesigned chromosomes, allowing the shuffling of Sc2.0 chromosome parts by the Cre-loxP recombination system thereby enabling genome evolution experiments. Here, we present L-SCRaMbLE, a light-controlled Cre recombinase for use in the yeast *Saccharomyces cerevisiae*. L-SCRaMbLE allows tight regulation of recombinase activity with up to 179-fold induction upon exposure to red light. The extent of recombination depends on induction time and concentration of the chromophore phycocyanobilin (PCB), which can be easily adjusted. The tool presented here provides improved recombination control over the previously reported estradiol-dependent SCRaMbLE induction system, mediating a larger variety of possible recombination events in SCRaMbLE-ing a reporter plasmid. Thereby, L-SCRaMbLE boosts the potential for further customization and provides a facile application for use in the *S. cerevisiae* genome re-engineering project Sc2.0 or in other recombination-based systems.

[1] Cell2Fab Research Unit, University of Potsdam, Karl-Liebknecht-Str. 24-25, 14476 Potsdam, Germany. [2] Institute for Systems Genetics, New York University Langone School of Medicine, New York City, NY 10016, USA. [3] Department of Molecular Biology, University of Potsdam, Karl-Liebknecht-Str. 24-25, 14476 Potsdam, Germany. [4] Max Planck Institute of Molecular Plant Physiology, Am Muehlenberg 1, 14476 Potsdam, Germany. [5]Present address: Department of Molecular Biology, University of Potsdam, Karl-Liebknecht-Str. 24-25, 14476 Potsdam, Germany. [6]Present address: Max Planck Institute of Molecular Plant Physiology, Am Muehlenberg 1, 14476 Potsdam, Germany. Correspondence and requests for materials should be addressed to B.M-R. (email: bmr@uni-potsdam.de)

The budding yeast *Saccharomyces cerevisiae* is one of the most intensively studied eukaryotic model organisms. It is a single cell organism with a relatively small (12 Mb) nuclear genome, has a short generation time, and can be easily cultured and genetically manipulated. Furthermore, it allows efficient protein expression with high productivity making it relevant for industry-scale production of proteins, enzymes and low-molecular-weight substances[1]. The yeast genome was the first eukaryotic genome to be sequenced in 1996[2]. The international Synthetic Yeast Genome Project, Sc2.0, aims to construct a redesigned, fully synthetic yeast genome; the re-engineering of the first 6.5 chromosomes has been reported[3–9]. An important feature of Sc2.0 chromosomes is Synthetic Chromosome Rearrangement and Modification by LoxPsym-mediated Evolution (SCRaMbLE), a global recombination system embedded in synthetic chromosomes during design to allow inducible genome evolution[10]. Briefly, loxPsym recombination sites are installed downstream of non-essential genes and at other major landmarks in the synthetic genome; once completed, the Sc2.0 genome will encode ~4000 loxPsym sites positioned across all sixteen synthetic chromosomes[11]. In contrast to natural loxP sites of the bacteriophage P1 Cre-loxP system, loxPsym sites contain a symmetric spacer region within the 34 bp long Cre recombination site[12–14]. In this way, Cre-mediated recombination between two loxPsym sites leads to deletions or inversions of DNA fragments, with theoretically equal probability[10,15,16]. SCRaMbLE is a useful tool for studying genome organization and stability, to develop new strains for improved heterologous production of small molecules and proteins, or to identify scrambled Sc2.0 derivative strains with reduced genome sizes. In the originally reported SCRaMbLE system, Cre is fused to the estrogen-binding domain (EBD) of the murine estrogen receptor[3,17]. Adding β-estradiol to the medium induces translocation of the Cre-EBD fusion protein from the cytosol to the nucleus, where Cre targets loxPsym sites. Although β-estradiol works as a robust inducer of SCRaMbLE, it has hormonal activity and can be toxic to humans at elevated concentrations, raising potential safety concern for the experimenter. Furthermore, shutting off Cre activity requires degradation of the Cre-EBD fusion protein and/or its dilution by cell division after the removal of β-estradiol from the culture medium.

Here, we present a light-controlled split Cre system that can be activated by red light in a dose-dependent and fully reversible manner. Our light-controlled induction system, dubbed L-SCRaMbLE, simplifies the tuning of SCRaMbLE dynamics and boosts the potential for further customization. L-SCRaMbLE is based on a split-protein design where the N- and C-termini of Cre (CreN and CreC, respectively)[18,19] are fused to two heterologous proteins, namely the chromophore-binding photoreceptor phytochrome B (PhyB) and its interacting factor PIF3 from the plant *Arabidopsis thaliana*[20,21]. The two plant proteins interact light-dependently to reconstitute a functional Cre recombinase in the presence of PCB, the chromophore of the PhyB photoreceptor. As a proof of concept, we scrambled a plasmid, harboring four loxPsym-flanked genes of the β-carotene biosynthesis pathway, in a light-controlled manner[22].

## Results

### Two tools for red light-regulated Cre-loxPsym-recombination.

L-SCRaMbLE combines an optical dimerizer from *Arabidopsis thaliana*[20] and a split Cre recombinase from bacteriophage P1[18,23], to be functional in the yeast *S. cerevisiae*. The light-sensitive system builds on an N-terminal version of the photoreceptor phytochrome B[24,25] (PhyBNT), which binds the chromophore phycocyanobilin (PCB)[26,27]. Upon red light illumination PhyBNT undergoes a conformational switch from

the Pr to the Pfr form, which is reversible by illumination with far-red light[28]. The Pfr form of PhyBNT binds to phytochrome interacting factor 3 (PIF3)[29,30]. The light-dependent switch of the PhyBNT-PIF3 interaction is employed here to allow light-controlled reconstitution of functional Cre recombinase from otherwise non-functional N- and C-terminal halves of the protein (split Cre recombinase). The N-terminal domain of the Cre protein (CreN) was fused to PhyBNT (creating PhyBNT-CreN protein), while its C-terminal domain (CreC) was linked to PIF3 (creating PIF3-CreC protein). The sequences of the split Cre recombinase and the linkers were adopted from a blue light-responsible split Cre based on the CRY2-CIB1 system from *A. thaliana* and reported to be functional in mammalian cells[18,19]. We tested two different light-dependent systems in this study, encoded by plasmids pLH_Scr15 and pLH_Scr16 (Fig. 1). Both systems harbor the nuclear localization signal of the simian virus 40 (SV40-NLS) within the PIF3-CreC fusion protein, but only pLH_Scr16 incorporates an additional NLS (from yeast Cdc1) within the PhyBNT-CreN fusion. In the pLH_Scr15 system, PIF3-CreC protein is expected to be nuclearly localized in the dark, while PhyBNT-CreN is predicted to be located in the cytoplasm. After red light illumination, PhyBNT-CreN binds PIF3-CreC and the protein complex formed moves to the nucleus (Fig. 1a). In the pLH_Scr16 system, both proteins, PIF3-CreC and PhyBNT-CreN are present in the nucleus in the dark. Upon red light illumination, PhyBNT-CreN binds PIF3-CreC which leads to reconstitution of functional Cre recombinase from the split halves (Fig. 1b). In both cases, reconstituted Cre initiates recombination at loxPsym sites present in the target DNA.

### L-SCRaMbLE recombination acts in a time-dependent manner.

To test L-SCRaMbLE for tunability of recombination frequency, we used the low-copy plasmid pLM494 (CEN/ARS), which harbors four functional genes of the β-carotene biosynthesis pathway each flanked by loxPsym sites (Supplementary Fig. 1). Yeast cells carrying the genes of this pathway produce β-carotene and its precursors resulting in yellow colonies, as previously shown for the β-carotene-producing parental plasmid pJC178 lacking the loxPsym sites[22]. In the presence of active Cre, recombination via the symmetric loxPsym sites can lead to diverse recombination events including inversions, deletions, translocations, and/or duplications of single or multiple loxPsym-flanked genes on pLM494[10,16]. While Cre-driven inversions of the loxPsym-flanked genes will likely not affect β-carotene production, deletion of *crtI* and/or *crtYB* is expected to result in a colorless product and, consequently, a white yeast colony[31]. Deletion of only *tHMG1* would cause orange colonies and deletion of *crtE* and *tHMG1* is expected to result in faintly yellow colonies[22,31]. Nevertheless, white colonies can be used to readout recombination events, and the ratio of white to yellow colonies to approximate recombination frequency. However, the fraction of white colonies observed post-SCRaMbLE induction is expected to be an underestimate for the following reasons: First, as CEN/ARS plasmids occur in 2–5 copies per cell[32] white colonies are only expected if all copies of pLM494 in a cell are recombined and second, the formation of dicentric plasmids due to recombination between two copies of pLM494 should cause cell death.

To test the efficiency of red light-inducible Cre, we transformed plasmid pLM494 together with plasmid pLH_Scr15 or pLH_Scr16 into yeast, or—to compare the light-inducible systems with the already established β-estradiol-inducible Cre system[3]—with plasmid pLM006. The resulting strains were designated Light-Cre1, Light-Cre2 and EST-Cre1, respectively. Light-Cre1, Light-Cre2 and EST-Cre1 cells were pre-cultured for 6 h in the dark. Subsequently, Light-Cre1 and Light-Cre2 cultures

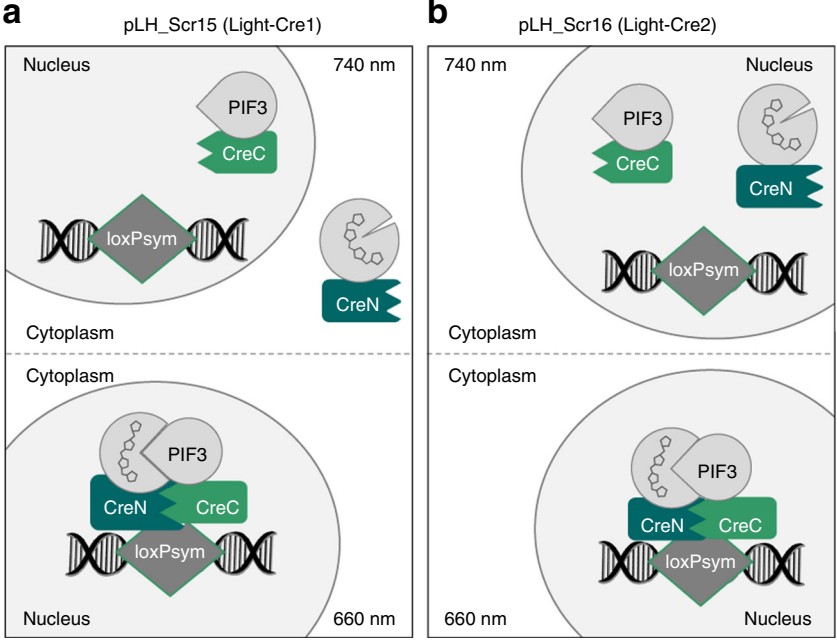

**Fig. 1** Schematic overview of L-SCRaMbLE. **a** Mode of action in cells harboring pLH_Scr15. In darkness or far-red light ($\lambda = 740$ nm), PIF3-CreC is located in the nucleus due to an incorporated NLS, while PhyBNT-CreN is located in the cytoplasm. Upon red light illumination ($\lambda = 660$ nm) PhyBNT-CreN binds PIF3-CreC in the cytoplasm (not shown), thereby reconstituting Cre recombinase from its split halves. Reconstituted Cre moves to the nucleus where it induces recombination between the loxPsym sites. **b** Mode of action in cells harboring pLH_Scr16. In darkness or far-red light ($\lambda = 740$ nm) PIF3-CreC and PhyBNT-CreN are separately located in the nucleus. Upon red light illumination ($\lambda = 660$ nm) PhyBNT-CreN binds PIF3-CreC, thereby reconstituting Cre recombinase from its split parts, allowing recombination at loxPsym sites

were induced with a red light pulse (660 nm) of 5 min, and grown for another 4 h or 16 h with 10-s red light pulses every 5 min. EST-Cre1 cultures were induced by adding 1 μM β-estradiol and subsequently grown for another 4 h or 16 h. Non-induced controls were kept in the dark (Light-Cre1/2) or in the absence of β-estradiol (EST-Cre1). Cells were then plated on appropriate selective media without PCB or β-estradiol and incubated for 2 days at 30 °C in the dark. For cultures plated after 4 h of induction, 14% and 27% of colonies from Light-Cre1 and Light-Cre2, respectively, were white, while 94% of EST-Cre1 colonies were white. In the non-induced samples, 2% and 10% of Light-Cre1 and Light-Cre2 colonies were white, compared to 40% of EST-Cre1 colonies (Fig. 2a; Supplementary Fig. 2A). For samples plated after 16 h of induction, 100% of the EST-Cre1 colonies were white (45% in non-induced samples). In the case of Light-Cre1 and Light-Cre2, 1% and 7% of the colonies, respectively, were white in non-induced cultures, similar to the 4 h experiment. However, the percentage of white colonies significantly increased in light-induced cultures (32% in Light-Cre1, 47% in Light-Cre2), consistent with ongoing light-dependent activation of L-SCRaMbLE-mediated recombination (Fig. 2b; Supplementary Fig. 2B). To raise the percentage of white colonies produced by L-SCRaMbLE even further, we extended the induction time to 24 h (Fig. 2c; Supplementary Fig. 2C). While the basal activity remained low in both systems (5% for Light-Cre1, 8% for Light-Cre2), the number of white colonies increased to 52% after red light induction in Light-Cre1 but remained largely unchanged (compared to the 16 h experiment) for Light-Cre2 (53% white colonies).

As a high basal activity of the EST-Cre system has not been reported within the context of a circular Sc2.0 chromosome arm, synIXR[33], which is ~90 kb long and encodes 43 loxPsym sites, we integrated two loxP sites, flanking an *URA3* cassette, into the *HO* locus of the yeast genome, resulting in strain yLM1295 (Supplementary Fig. 3A). After transformation with plasmid pLM006,

the newly generated strain EST-Cre2 was tested with regard to the recombination activity between two loxP sites after β-estradiol induction. As successful recombination will cause deletion of the *URA3* cassette, recombination activity was evaluated by plating samples on synthetic complete (SC) drop-out media with or without 5-fluoroorotic acid (FOA) for counter selection of *URA3* expression (Supplementary Fig. 3A). While cells with a successfully deleted *URA3* gene grow on FOA-containing medium, cells expressing the *URA3* gene do not. Estradiol-induced samples showed recombination efficiencies of only 6.5% after 24 h of induction, while uninduced samples showed 1.4% background recombination (Supplementary Fig. 3B and C). This demonstrates that Cre recombinase acts less efficiently on two genome-integrated loxP sites than on multiple episomal loxPsym sites (100% recombination activity after already 16 h incubation for induced samples, and 45% for uninduced ones, Fig. 2). As recombination events between two loxP or two loxPsym sites were shown to occur with approximately equal efficiency, at least in *Escherichia coli* cells, we assume that the observed difference in recombination efficiency is not due to the spacer region within the recombination sites[15]. As the observed recombination efficiencies of the light-inducible systems Light-Cre1 and Light-Cre2 were up to 20 times lower than in the β-estradiol inducible system EST-Cre1, when acting on pLM494, the number of FOA resistant clones generated in the context of two genome-integrated loxP sites was too low for a reliable evaluation of recombination efficiency.

**L-SCRaMbLE mediates diverse recombination outcomes**. As diverse recombination events are desirable for SCRaMbLE-ing a synthetic yeast genome, we investigated recombination outcome mediated by EST-Cre1 and Light-Cre1. To this end, we isolated plasmids from white and yellow colonies from induced and non-induced plates of the 16 h experiment. Plasmids were

retransformed into *E. coli*, isolated after culture, and digested with restriction enzymes *Pst*I and *Sac*I. The resulting fragments were analyzed to identify individual recombination events. The restriction patterns were matched to those of pLM494 and different hypothetical plasmids resulting from possible recombination events. One representative plasmid of each pattern was sequenced. For Light-Cre1, we isolated plasmids from two white and two yellow colonies from a non-induced plate. As expected, restriction patterns matched those of unmodified pLM494 for the yellow colonies, and different gene deletions for the white colonies. This result is consistent with the assumption that yellow colonies carry genetically intact β-carotene pathways, while white colonies contain recombined plasmids with gene deletions dependent on basal Cre activity since all rearrangements occurred specifically at loxPsym sites (Fig. 3). The patterns of 12 plasmids from light-induced colonies (2 yellow, 10 white) showed eight different recombination events, all of which differed from that of pLM494. This demonstrates that L-SCRaMbLE yields a wide spectrum of unique deletion and inversion events after light induction. Moreover, the fact that most plasmids yielded different digestion patterns is consistent with the recovery of plasmids from unrelated yeast cells, indicating that the 16 h induction culture is not dominated by siblings carrying recombined pathways from a small number of mother cells. This is most likely due

to the repeated light pulses that were applied over the whole induction period. We did not observe any restriction patterns consistent with duplication events in this analysis, possibly because we focused largely on studying events in white colonies post-induction. Plasmids recovered from two yellow colonies post-light induction unexpectedly had restriction patterns consistent with deletion of β-carotene pathway genes (Fig. 3, plasmids 5 and 6). Specifically, digestion of plasmid 6 suggests deletion of both *tHMG1* and *crtYB* genes and plasmid 5 is consistent with deletion of all four pathway genes; both plasmids should yield white colonies. We hypothesized that the original yeast colonies encoded multiple uniquely recombined pathways or that we had recovered the plasmid from a mixed colony. To investigate this further, four additional independent *E. coli* colonies for both plasmids 5 and 6 were analyzed by DNA restriction. The restriction patterns confirmed that both parental colonies contain a mixture of different plasmids (Supplementary Fig. 4A). While plasmids originating from colony 6 collectively showed to contain all four genes required for β-carotene biosynthesis, the plasmids isolated from colony 5 are expected to yield a white colony. To distinguish whether the parental yeast colony contained a mixture of different cells or whether cells of the parental colony are identical but contain multiple variants of the β-carotene plasmids, the cells were streaked out to obtain single

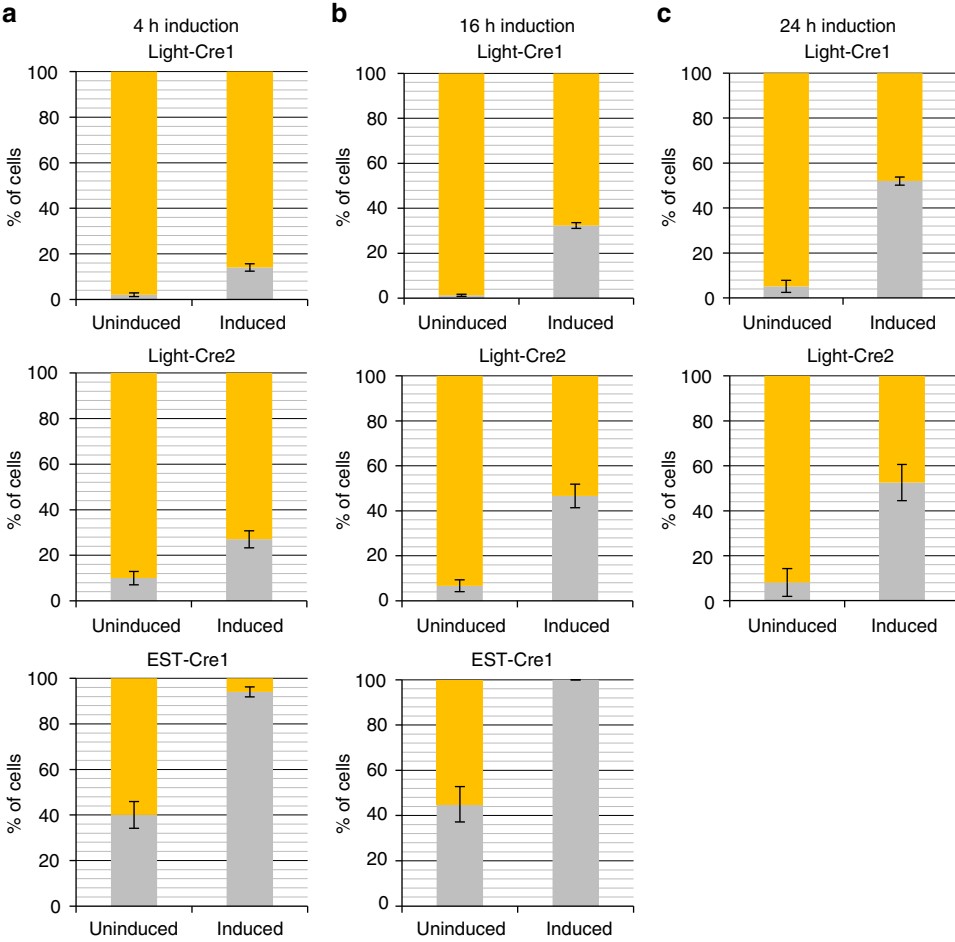

**Fig. 2** Recombination experiment with incubation times of 4 h, 16 h or 24 h. Light-Cre1, Light-Cre2 and EST-Cre1 cells were grown for 6 h in darkness. Light-Cre1 and Light-Cre2 cells were grown in media containing 25 μM PCB. After induction with a 5-min red light pulse or by addition of β-estradiol (1 μM final concentration), samples were grown for further **a** 4 h, **b** 16 h, and **c** 24 h at 30 °C and 230 rpm. One ml of each culture was pelleted, diluted and plated on appropriate synthetic complete (SC) drop-out media. Each experiment was performed in three independent replicates. Bar blots show the mean values of the percentages of white and yellow colonies in induced and uninduced samples for strains Light-Cre1, Light-Cre2 and EST-Cre1. Grey, white colonies; yellow, yellow colonies. Error bars indicate standard deviation (SD)

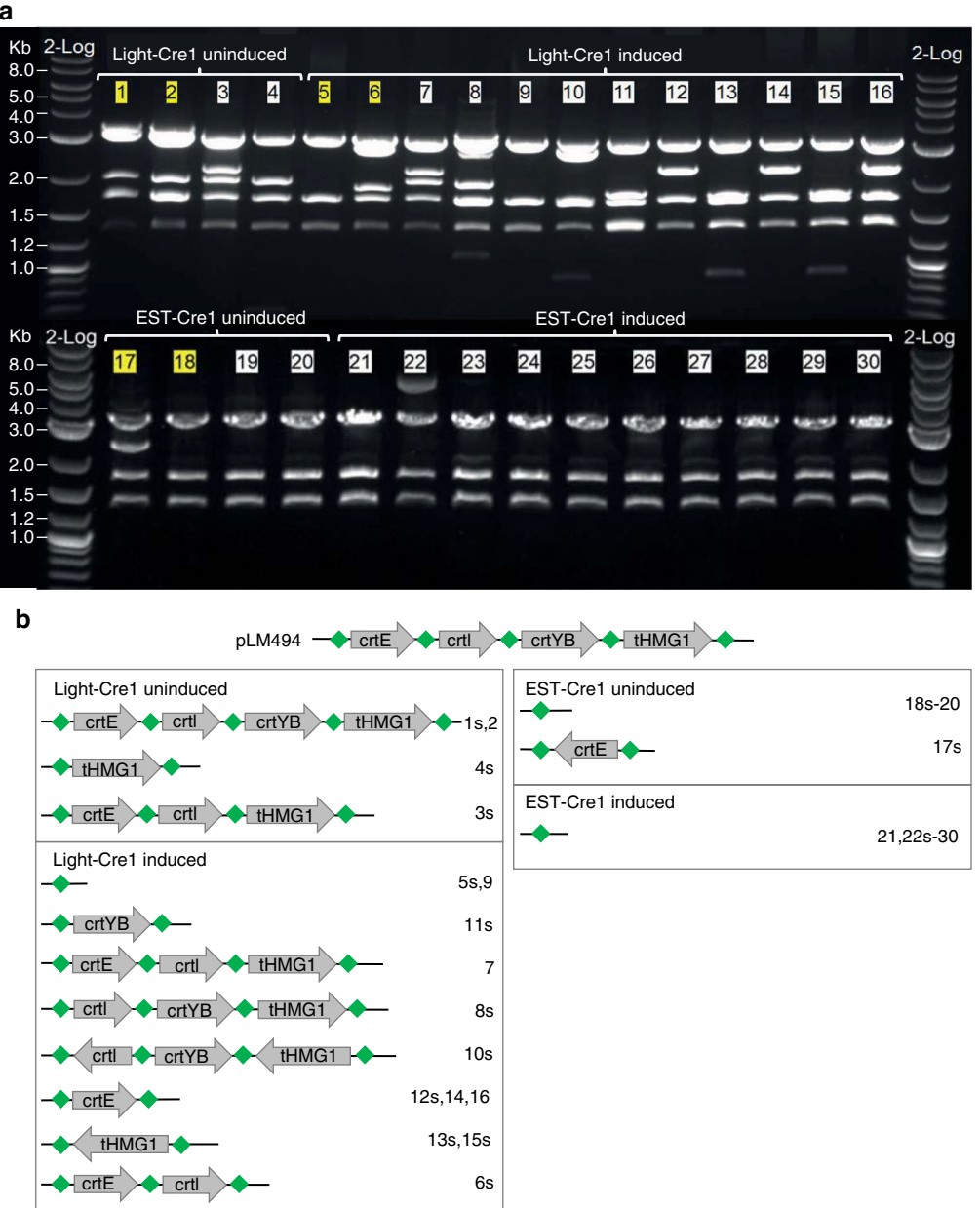

**Fig. 3** Restriction patterns of pLM494 after Cre-mediated recombination. Plasmids were isolated from different yeast colonies of induced and uninduced Light-Cre1 and EST-Cre1 cultures (16 h) and after passage through *E. coli* digested with *Pst*I and *Sac*I. **a** Gel electrophoresis of digested plasmids. Numbers shaded in yellow show patterns of plasmids isolated from yellow yeast colonies, while numbers shaded in white show patterns of plasmids isolated from white colonies. ´2-Log´, DNA Ladder (NEB). Kb: Kilobases. **b** Overview of the recombination events. Gene names are given. Numbers on the right refer to the corresponding plasmid restrictions shown in **a**. ´s´ indicates sequence-verified plasmids. Green diamonds indicate loxPsym sites

colonies (Supplementary Fig. 4B). For colony 5 we observed a mixed population of white and yellow colonies. Plasmids of a white and yellow yeast colony were isolated and retransformed into *E. coli*, and four independent *E. coli* colonies were tested by restriction analysis, confirming uniform plasmid population for the white colony. For the yellow colony, a mixture of three plasmids with a single *crtI* deletion and one plasmid with a *crtE* inversion was obtained. Both results fit the observed colony color (Supplementary Fig. 4C). For colony 6, almost no white colonies were detected upon re-streaking, suggesting cells of colony 6 contained different β-carotene plasmids.

As both investigated yellow colonies obtained after induction showed recombined plasmids, we isolated plasmids from 18 additional yellow yeast colonies to check their restriction patterns

(Supplementary Fig. 5). Remarkably, all colonies contained recombined plasmids, which was confirmed by sequencing in cases where digestion patterns were inconclusive (plasmids 33 and 36). Again, we observed a large diversity among the recombination patterns.

For EST-Cre1, we also investigated plasmids isolated from four non-induced colonies (2 yellow, 2 white) and found that none of them showed restriction patterns matching pLM494 (Fig. 3). This indicates that background recombination of EST-Cre1 in uninduced conditions is even higher than the one calculated from the ratio of white to yellow colonies (Supplementary Fig. 6). Sequencing confirmed that plasmid 17 contained an inverted *crtI* sequence, while plasmids 18, 19, and 20 showed recombined pLM494 with deletions of all four β-carotene biosynthesis genes.

All 10 plasmids isolated from white colonies of the EST-Cre1 induced experiment (16 h) also showed restriction patterns of recombined pLM494 with deletions of always all four genes.

**L-SCRaMbLE allows PCB-dependent fine-tuning of recombination**. As L-SCRaMbLE allows light-triggered recombination to be controlled in a time-dependent manner, we next investigated whether PCB-dependent regulation is also feasible. PCB, the chromophore of the PhyB photoreceptor, enables the light-responsive conformational switch of the holoenzyme from the Pr to the Pfr form. PCB dose-dependency was previously reported for a gene expression control system in *S. cerevisiae*[34]. Recently, we confirmed this result by employing the red light-regulated gene expression system PhiReX, which builds on the same optical dimerizer[21]. In the study presented here, the concentration of PCB supplied to SC-Ura-Leu media was varied from 0 to 100 μM. Light-Cre1 cells were pre-cultured for 6 h in the dark, subsequently induced with a red light pulse of 5 min, and grown for another 16 h with 10-s red light pulses every 5 min before plating. Non-induced samples were grown in the dark throughout. With this approach, the percentage of white colonies can be varied stepwise from 0.5% in the absence of PCB to 90% in the presence of PCB with concentrations of up to 100 μM in the medium (Fig. 4 and Supplementary Fig. 7). With PCB concentrations of up to 25 μM, cultures grown in the dark always yielded fewer than 1% white colonies; these values increased up to 10% with PCB concentrations of 100 μM. Nevertheless, in a SCRaMbLE experiment, the user would only add PCB to the medium when recombination is intended. In this way, L-SCRaMbLE guarantees reliable off-states with < 1% recombination in the absence of PCB, while PCB levels can be used to fine tune recombination efficiency.

**L-SCRaMbLE mediates reliable off-states**. For SCRaMbLE experiments, controlled termination of Cre activity is an important aspect to ensure stability of the newly generated genetic construct or genome. The results shown here demonstrate that virtually no recombination events occur in the absence of the chromophore PCB, regardless of the light condition used. This suggests that Cre can be inactivated by transferring cells to medium lacking PCB, in addition to inactivation by far-red light application. To confirm this, 10 yellow and 10 white colonies resulting from samples induced in medium with 25 μM PCB (Fig. 4c) were re-streaked on PCB-free medium (Supplementary Fig. 8A, B). After re-streaking, all 10 white colonies produced white sibling colonies, as expected. Of the 10 yellow colonies, eight produced sibling colonies of identical color as the parental ones. Only two colonies (numbers 4 and 8; Supplementary Fig. 8A) resulted in very few white colonies, while the vast majority of sibling colonies retained the same color as the parental ones. The plate assay thus indicated low Cre recombination activity after removal of PCB from the medium. To verify this, plasmids were isolated from four single colonies derived from two yellow (A1 and A5, Supplementary Fig. 8A) and two white parental colonies (B2 and B5, Supplementary Fig. 8B) and retransformed into *E. coli*. Plasmids extracted from four independent *E. coli* colonies were analyzed by enzyme restriction. All 16 plasmids isolated from four sibling colonies of parental colony B5 showed deletions of all four pathway genes. Consequently, these plasmids cannot be used to monitor undesired recombination events in the off-state. However, also all 16 plasmids derived from four sibling colonies of parental colony B2 harbored the same plasmid (*tHMG1* only) without further recombination events (Supplementary Fig. 8C). For each of the yellow parental colonies A1 and A5, 15 out of 16 plasmids passaged through *E. coli* showed

restriction patterns matching those of plasmid pLM494. Only one plasmid from A1 (colony 3_2) and A5 (colony 4_4) differ from the parental colony resulting from gene deletions.

**Recombination efficiency towards a loxP-flanked DNA fragment**. By evaluating L-SCRaMbLE activity utilizing plasmid pLM494 as the template we gained in-depth insight into the tunability of recombination activity and the diversity of recombination products. However, as outlined before, absolute recombination efficiency cannot be precisely judged in this way due to unknown and likely diverse plasmid copy numbers in each cell and the fact that some recombination events, like e.g. inversion and duplications, cannot be easily detected by restriction enzyme digests. To more precisely determine the efficiency of L-SCRaMbLE to trigger individual recombination events we therefore constructed plasmid pLH_Scr18, which contains a loxP-flanked double terminator cassette between an *ADH1* promoter and the CDS of yEGFP, which prevents yEGFP expression (Fig. 5a). Only upon recombination between the two loxP sites the terminators are deleted, resulting in yEGFP fluorescence. Of note, we chose loxP rather than loxPsym sites to prevent mere inversion of the terminator cassette. Using the β–estradiol inducible system, dubbed EST-Cre3, we measured 77% yEGFP-positive cells upon 24 h induction with 1 μM β–estradiol, and 11% yEGFP-positive cells for uninduced cells (Fig. 5b). We next tested L-SCRaMbLE on plasmid pLH_Scr18 (Light-Cre3), which carries a loxP-flanked terminator cassette in between the constitutive *ADH1* promoter and the CDS of yEGFP. After 24 h red light induction, 12% yEGFP-positive cells were observed, while only 0.6% of the untreated cells showed yEGFP fluorescence above the measurable threshold. Cells harboring plasmid pLH_Scr19 (Fig. 5a) were used as positive controls where yEGFP signal in a large fraction of the cells is expected in the absence of any recombination. The results obtained reflect the differences in recombination activity between L-SCRaMbLE and EST-Cre, already observed when testing both systems on plasmid pLM494. The generally lower recombination activity of L-SCRaMbLE compared to SCRaMbLE explains the wanted large diversity of recombination products when acting on several symmetrical loxPsym sites.

## Discussion

In this study, we demonstrate the development of a light-regulated Cre recombinase for application in the yeast *S. cerevisiae*. We established two systems, Light-Cre1 (pLH_Scr15) and Light-Cre2 (pLH_Scr16), both of which are based on red light-mediated dimerization of the plant photoreceptor PhyB and its interacting factor PIF3. By fusing the N- and C-terminal halves of a split Cre recombinase to the two proteins of the optical dimerizer, recombination activity can be well controlled in a light-dependent manner (Fig. 1). Both systems were compared to the previously reported β-estradiol inducible Cre-EBD system (EST-Cre1, pLM006)[10]. Plasmid pLM494, which contains four loxPsym-flanked genes of the β-carotene pathway and produces yeast yellow colonies if no deletion/recombination of individual genes occurs[22], was used to assess tunability of recombinase activity and diversity of recombination products. Both criteria are important factors for SCRaMbLE-ing synthetic (Sc2.0) yeast chromosomes.

The data obtained here reveal L-SCRaMbLE as a reliable tool with very low basal activity in the absence of light and high recombination capacity after red light exposure. Importantly, recombination frequency can be fine-tuned in a PCB-, time- and light dose-dependent fashion.

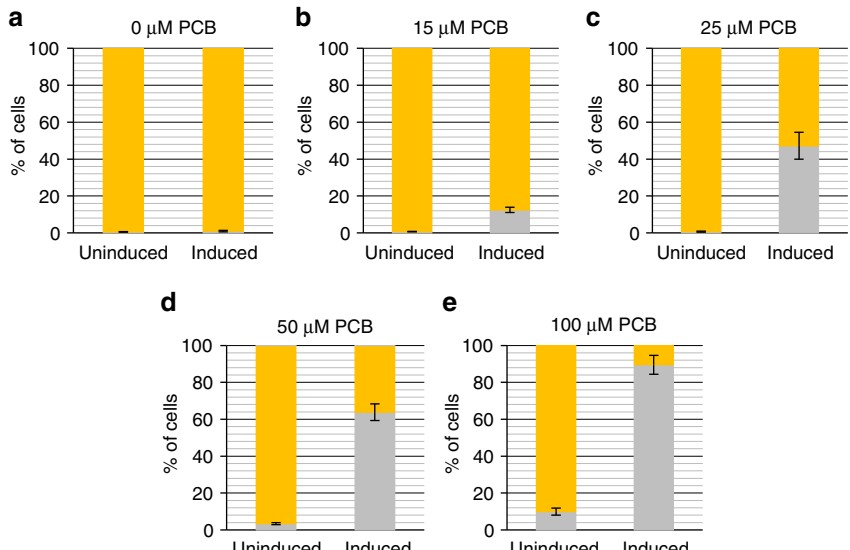

**Fig. 4** L-SCRaMbLE-mediated recombination at varying PCB concentrations. Light-Cre1 cells were grown in SC-Ura-Leu media with **a** 0, **b** 15, **c** 25, **d** 50 and **e** 100 μM PCB for 6 h in darkness. Induced cells were treated with a 5-min red light pulse, followed by 16 h growth with 10-s red light pulses every 5 min. Uninduced samples were grown for 16 h in darkness. Around 100 μl of each culture were pelleted, diluted and plated on SC-Ura-Leu. Bar blots show the means of the percentages of white and yellow colonies in induced and uninduced samples. Grey, white colonies; yellow, yellow colonies. Data are the means of three independent replications ± SD

Time-dependent recombination was more pronounced in Light-Cre1 (three-fold more recombination events after 24 h than 4 h) than Light-Cre2 (two-fold increase of recombination after 24 h compared to 4 h). Light-Cre1 also exhibited lower basal activity than Light-Cre2, consistent with the fact that in Light-Cre1 the PIF3-CreC fusion protein resides in the nucleus, while PhyBNT-CreN is located in the cytosol before induction, thereby avoiding spurious reconstitution of the split Cre recombinase (Fig. 1). In view of these results we employed plasmid pLH_Scr15 in further experiments and showed that chromophore concentration can be used to control recombination efficiency from 0.5% (in the absence of PCB) up to 90% (with 100 μM PCB in the medium; Fig. 4), resulting in a 179-fold increase in observed recombination events. Although basal recombination levels increased with higher PCB concentration, L-SCRaMbLE can be utilized for tight regulation of Cre activity; to this end the experimenter can transfer the synthetic yeast strain into PCB-containing medium only when recombination is desired. When maintaining the synthetic yeast strain in PCB-free medium, background recombination is negligible (below 1%; Supplementary Fig. 7) and independent of the light condition. Furthermore, by withdrawal of PCB, Cre recombinase activity can be completely shut down. This guarantees the formation of isogenic yeast colonies after stopping the SCRaMbLE experiment (Supplementary Fig. 7).

The Sc2.0 project incorporates SCRaMbLE as an inducible system to investigate genome evolution-related aspects and to generate reduced or even minimal genomes. Beneficial in such studies are numerous small-scale modifications of the native genome, allowing the investigation of many different genomic states, while large deletions encompassing essential genes are expected to cause cell death. We therefore had a deeper look at the rearrangements to the β-carotene pathway caused by Light-Cre1 and EST-Cre1 (Fig. 3, Supplementary Fig. 5). Light-Cre1 induces a large variation of recombination events. Eight out of twelve initially investigated plasmids (isolated from two yellow and ten white colonies) showed different plasmid rearrangements (Fig. 3). Furthermore, 18 plasmids additionally isolated from yellow colonies of induced samples show (mostly different) recombination events (Supplementary Fig. 5), confirming the assumption that the recombination activity is considerably higher than judged from the ratio of white to yellow colonies. In contrast, EST-Cre1 always results in the same plasmid with deletions of all four biosynthesis genes. Notably, each of the four EST-Cre1 colonies of the uninduced experiment contained recombined plasmids, indicating that the basal activity is even higher than expected from counting white colonies.

The L-SCRaMbLE system has great advantages over the chemically inducible EST-Cre system utilized in SCRaMbLE experiments so far. Although β-estradiol-induced Cre recombinase demonstrates highly efficient recombination, it becomes obvious that EST-Cre cannot be regulated in a time-dependent manner and exhibits high basal activity in the absence of inducer (Fig. 2). Even though an additional EBD fused to the Cre-EBD protein appears to minimize background activity in a β-estradiol-inducible system[35], fine-tuning and a reversible control of Cre recombinase cannot be easily achieved. The differences between Light-Cre and EST-Cre with respect to recombination efficiency and background activity were also observed in an unbiased test system containing two loxP sites flanking a deletable DNA fragment (Fig. 5). As for experiments with pLM494, we observed a much higher recombination background, but also a higher inducible recombination activity, for EST-Cre than for Light-Cre. We assume that the overall lower recombination efficiency of Cre, when applied on plasmid pLH_Scr18 (containing only two loxP sites), compared to plasmid pLM494 (carrying five loxPsym sites) results from the available number of recombination sites. Different recombination efficiencies due to the usage of loxP in contrast to loxPsym sites are unlikely, as Cre recombinase was previously shown to act with similar efficiencies on these two recombination sites[15]. In the context of SCRaMbLE experiments with synthetic Sc2.0 chromosomes it becomes obvious that lower recombination activity favors more diverse recombination outcomes, which is expected to be highly advantageous when performing genome evolution experiments. Also, the very low background activity mediated by L-SCRaMbLE is an important asset for such experiments as it favors the generation of isogenic yeast strains.

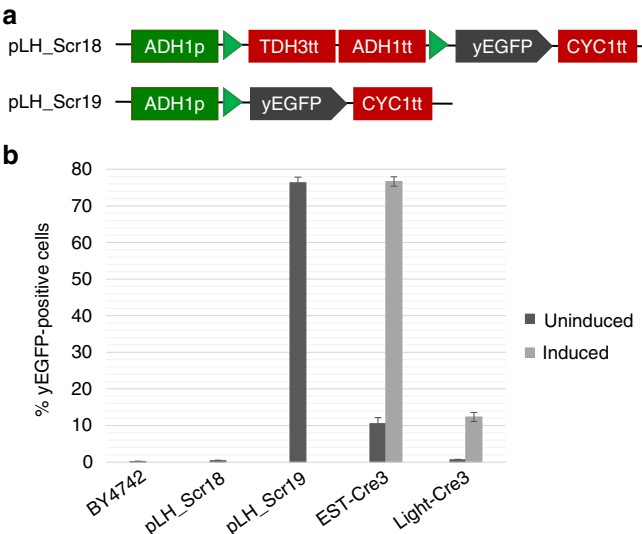

**Fig. 5** Measurement of recombination efficiency. **a** Recombination efficiency was determined using plasmid pLH_Scr18, which contains a loxP-flanked double terminator cassette upstream of the yEGFP CDS. Recombination between the loxP sites, shown as green triangles, results in deletion of the terminators allowing yEGFP expression. Plasmid pLH_Scr19 served as positive control. **b** Wild-type and pLH_Scr18-transformed BY4742 cells served as negative controls for yEGFP signal, while BY4742 cells containing plasmid pLH_Scr19 served as positive control. EST-Cre3 cells contain plasmids pLH_Scr18 and pLM006, while Light-Cre3 cells contain pLH_Scr18 and pLH_Scr15. After 6 h of preculture, EST-Cre3 cells were induced with 1 µM β-estradiol, and Light-Cre3 cells with red light pulses. After 24 h of induction the number of yEGFP-positive cells was determined by flow cytometry. Uninduced samples as well as controls were grown in darkness. Data are the means of three independent replications ± SD

Remarkably, the high basal activity of the β-estradiol inducible system has not been observed in the context of the circular Sc2.0 chromosome arm synIXR[33] or when applying the β-estradiol inducible pLM006 system on two single genome-integrated loxP sites (Supplementary Fig. 3). In the latter experiment, the recombination efficiencies in induced and uninduced samples were much lower than the recombination efficiencies of EST-Cre while acting on plasmid pLM494 or pLH_Scr18. However, as compared to recombination efficiency of 6.5% after induction, EST-Cre2 still shows a high background activity of 1.4%. Our results clearly demonstrate, that Cre recombinase acts less efficiently on genome-integrated loxP sites compared to episomal ones (Figs. 2, 5b). This may explain why Shen et al. reported no background recombination with synthetic yeast genome synIXR[33]. Notably, synIXR contains only 43 loxPsym sites, while the final synthetic yeast genome will harbor 4000 loxPsym sites. This makes background recombination much more likely to occur in the final (complete) synthetic yeast genome and highlights the advantage of L-SCRaMbLE.

While L-SCRaMbLE is the first light-regulated Cre recombinase for use in yeast, several blue light-responsive systems are available for mammalian cells that are based on plant-derived CRY2-CIB1 proteins[18,19] or the Magnet system, engineered from a fungal photoreceptor[36]. Regarding the fold-induction or the percentage of recombination activity reported for those systems, L-SCRaMbLE shows comparable, or even better results depending on the reporter system. However, while single blue light pulses are sufficient to activate Cre recombinase in mammalian systems, L-SCRaMbLE depends on repetitive red light pulses applied over a longer period. Although this may appear

unfavorable at first glance, a red light-controlled recombination system bears considerable advantages in yeast. First, the current blue light-inducible systems depend on a chromophore naturally present in yeast, precluding fine-tuning of recombinase activity by varying chromophore concentration. Second, a complete shut-off, independent of light status, is not possible. Third, blue light affects yeast growth and metabolism, while red light only negligibly affects yeast physiology[37]. This eliminates the need for a system that fully responds to a single red light pulse. Fourth, multi-pulse induction over several hours is highly beneficial for a light-inducible SCRaMbLE system. While short pulses are advantageous for applications in mammalian cells, induction with a single pulse in yeast would result in few recombination events that are genetically inherited over the time span of the induction experiment. When a single pulse is applied to the samples, PhyB will dimerize with PIF3 and the functionally reconstituted Cre recombinase can act on loxPsym sites. However, after some time in darkness, PhyB will return to its Pr state and the PhyB-PIF3 dimer will dissociate, leaving Cre recombinase non-functional. Thereafter, recombination events will be inherited approximately every 1.5 h due to cell division, without creating new recombinations. Naturally, photoreceptor proteins expressed after single pulse induction will also be inactive. As generation times of mammalian cells are much longer compared to yeast, this effect might be negligible in those systems. Furthermore, while the tools established for mammalian cells intended to generate single specific recombination outcomes, SCRaMbLE-ing a synthetic yeast genome aims at generating numerous independent and diverse recombination events. This can only be achieved with moderate recombination activity acting over various periods of time, as made possible with L-SCRaMbLE reported here.

Taken together, L-SCRaMbLE allows for tight control of recombination pre-induction and high Cre-mediated recombination after light induction. The fact that L-SCRaMbLE can additionally be controlled by PCB concentration makes it an excellent tool for fine-tuning recombination frequency in yeast, according to experimental needs. Furthermore, withdrawal of PCB from the medium allows a complete shut-down of L-SCRaMbLE-mediated recombination. When applied to Sc2.0 synthetic strains[4–9], which will harbor thousands of loxPsym sequences across all synthetic chromosomes, L-SCRaMbLE will serve as an excellent tool for an exquisite control over Cre activity.

## Methods

**Strains and reagents**. *E. coli* strain DH5α was used for cloning purposes. For experiments with *S. cerevisiae*, strain BY4742 was cultured at 30 °C in Yeast extract Peptone Dextrose Adenine (YPDA)-rich medium or in appropriate SC media lacking one or more amino acids to select for the maintenance of transformed plasmids. NucleoSpin Gel and PCR Clean-up kit (Macherey Nagel, Düren, Germany) were used for PCR purification and gel elution, and plasmid purification was done with NucleoSpin Plasmid EasyPure kit (Macherey Nagel).

**Cloning and DNA manipulation techniques**. Restriction-ligation cloning was carried out according to standard procedures. Several different overlap-directed DNA assembly methods were used to produce constructs described in this work, including SLiCE cloning using DH10B lysate[38,39], in vivo homologous recombination in yeast, and Gibson assembly[40]. Adjacent pieces for assembly were produced by PCR using Phusion polymerase (Life Technologies GmbH, Darmstadt, Germany) to encode 20–40 bp of terminal homology. Correct assemblies were verified by DNA sequencing. For dephosphorylation of digested vector backbones Antarctic Phosphatase (New England Biolabs, Frankfurt am Main, Germany) was used. Restriction enzymes were purchased from New England Biolabs.

**Constructs and strains**. Plasmid pLM006 was made by Gibson assembly, amplifying the pSCW11-Cre-EBD transcription unit from pDL12[17] and inserting it into the *Sma*I site of pRS413[41]. Plasmid pLM494 was built using a previously assembled construct encoding the β-carotene pathway, pJC178, as starting material[22]. Briefly, PCR amplicons spanning each of the four β-carotene transcription

units (TU) were generated using overhang primers to add a loxPsym sequence to one end of each TU and BsaI sites to both ends. The BsaI sites were oriented to cut themselves out, leaving non-palindromic, unique 4-bp overhangs for directional assembly with a 'yeast Golden Gate' acceptor vector[42]. To carry out type IIS restriction enzyme-based Golden Gate reactions, 100 ng of Golden Gate acceptor vector plus equimolar amounts of PCR-amplified DNA fragments were mixed together for one-pot digestion-ligation reaction. Each reaction included 1.5 μl 10x T4 DNA ligase reaction buffer (New England Biolabs), 0.15 μl 100x bovine serum albumin (New England Biolabs), 600 U T4 DNA ligase (Qiagen, Hilden, Germany), and 10 U of BsaI (New England Biolabs) in a final volume of 15 μl[42].

Plasmids pLH_Scr15 and pLH_Scr16 were constructed with the overlap-based DNA assembly methods SLiCE or TAR cloning and by using the AssemblX toolkit[38,43,44]. To this end, pLH_Scr12, pLH_Scr13 and pLH_Scr14 were constructed as precursor (Level 0) plasmids according to the AssemblX cloning procedure[43]. All parts used in this study to clone pLH_Scr12 to pLH_Scr16, the appropriate sources and sequence modifications, if applicable, are listed in Supplementary Table 1, while sequences of PCR primers are provided in Supplementary Table 2. Sequences and plasmid maps of all constructs described here are given in Supplementary Data File 1 and Supplementary Figs. 9, 10 and 11.

To construct Level 0 plasmids pLH_Scr12 and pLH_Scr13, plasmid pRot4 containing a fusion of an N-terminal version of PhyB (amino acids 1-621; PhyBNT) and the nuclear localization signal Cdc1-NLS under control of the TDH3 promoter and TDH3 terminator was linearized by BamHI/AgeI restriction. Subsequently, the DNA fragments encoding CreN, resulting from PCR on pLM006 with primers L714/L777 and L777/L778, respectively, were inserted via SLiCE cloning. We mutated a few nucleotides in the CreC encoding sequence to match the CreC amino acid sequence encoded by plasmid pmCherry-CIBN-CreC[18] (see Supplementary Table 1). A linker from plasmid pmCherry-CRY2-CreN was used to fuse CreN to PhyBNT, and CreC to PIF3, respectively[18]. To construct pLH_Scr14, plasmid pRot1, containing the yeast FBA1 promoter and terminator, was linearized with NotI. Thereafter, a PCR fragment encoding the PIF3-SV40NLS fusion was generated with primers L280/L773 and inserted into pRot2. A DNA fragment encoding CreC was obtained by PCR on template pLM006, with primers L712/L774 and L775/L776. All three PCR products were inserted via SLiCE cloning into linearized pRot1 backbone. To construct Level 1 plasmid pLH_Scr15, Level 0 plasmids pLH_Scr12 and pLH_Scr14 were digested with PmeI and the resulting fragments containing homology regions A0/A1 and A1/A2, respectively, were combined together with annealed oligonucleotides L706/L707 into PacI-linearized pL1A0*B0*_Leu[43] via TAR cloning. To construct Level 1 plasmid pLH_Scr16, Level 0 plasmids pLH_Scr13 and pLH_Scr14 were digested with PmeI and the resulting large fragments containing homology regions A0/A1 and A1/A2, respectively, were combined together with annealed oligonucleotides L706/L707 into PacI-linearized pL1A0*B0*_Leu[43] via TAR cloning.

To construct plasmid pLH_Scr18, three PCR-amplified DNA fragments were assembled. The first fragment resulted from amplification of pRL_yEGFP_ADH1p_hc[21] with primers L813/L802, the second from amplification of the TDH3 terminator from plasmid pPC_011[43] with primers L814/L815, and the third fragment from amplification of the ADH1 terminator from plasmid pPC_001[43] with primers L816/L817. To construct pLH_Scr19, two PCR fragments resulting from amplification of pRL_yEGFP_ADH1p_hc[21] with primers L802/L803 and L822/L823, respectively, were assembled.

Plasmids pLH_Scr15, pLH_Scr16 and pLM006, respectively, were cotransformed with plasmid pLM494 into BY4742 to yield strains Light-Cre1, Light-Cre2 and EST-Cre1. Plasmid pLH_Scr18 was cotransformed into BY4742 with plasmids pLM006 and pLH_Scr15, respectively, to yield strains EST-Cre3 and Light-Cre3. Plasmids pLH_Scr18 and pLH_Scr19 were separately transformed into BY4742 as a negative and positive control, respectively.

Yeast strain yLM1295 was built by transforming a PCR product derived from plasmid pUG72[45] (Euroscarf #P30117), generated with overhang primers targeting the HO locus for integration. Following selection on SC–Ura, integrants were confirmed by PCR.

Plasmid pLM006 was transformed into yLM1295, resulting in yeast strain EST-Cre2. S. cerevisiae transformations were carried out using the LiAc/SS carrier DNA/PEG method by Gietz and Schiestl[46].

Plasmid sequences are available in Supplementary Data 1–13.

**Recombination experiments**. For recombination experiments, Light-Cre1, Light-Cre2, and EST-Cre1 cells were inoculated in 3 ml appropriate drop-out medium and cultured overnight at 30 °C on a rotary shaker (230 rpm). Hundred μl of overnight grown Light-Cre1 and Light-Cre2 cultures were inoculated in 10 ml SC-Ura-Leu medium containing 25 μM PCB (Livchem, Frankfurt am Main, Germany) in 100-ml baffled Erlenmeyer flasks. Similarly, 100 μl of EST-Cre1 overnight culture were inoculated in 10 ml SC-Ura-His medium in 100-ml baffled flasks. For inactivation, Light-Cre1 and Light-Cre2 cultures were irradiated with a 1-min far-red light pulse and afterwards grown for 6 h in darkness at 30 °C and 230 rpm. Thereafter, cultures were irradiated with a 5-min red light pulse and then grown for 4, 16, or 24 h with 10-s red light pulses applied every 5 min. Non-induced samples were maintained in darkness throughout. EST-Cre1 cells were grown in darkness for 6 h at 30 °C and 230 rpm, then induced with 1 μM β-estradiol (Sigma-Aldrich, Munich, Germany) and grown for 4 or 16 h. Non-induced samples were

maintained in medium without β-estradiol. For all samples, 1 ml of cells was pelleted after 4 h incubation and 100 μl of cells were pelleted after 16 h and 24 h incubation, redissolved in 1 ml ddH$_2$O, diluted 1:100 and 1:1000 in ddH$_2$O, and plated on SC-Ura-Leu (Light-Cre1/2/3) or SC-Ura-His (EST-Cre1/3).

To test PCB-dependent recombination, overnight and main cultures of strain Light-Cre1 were inoculated and treated as described above, except that PCB concentration was varied (0, 15, 25, 50, and 100 μM) in the 10-ml SC-Ura-Leu media.

Main cultures of EST-Cre2 cells were inoculated and induced in SC-His medium, as described above. For all samples, 100 μl of cells were pelleted after 24 h incubation, redissolved in 1 ml ddH$_2$O, diluted 1:100 in ddH$_2$O, and plated on SC-His medium containing 1 mg ml$^{-1}$ FOA.

**Analysis of recombination experiments**. For experiments carried out with Light-Cre1, Light-Cre2, and EST-Cre1 cells, recombination efficiency was calculated by the ratio of white to yellow colonies. To this end, white and yellow colonies of one plate of each replicate were counted. Orange or slightly yellow colonies were counted as yellow colonies. Mixed colonies showing white and yellow sectors were counted as white, as well as yellow colonies. For experiments carried out with yeast strain EST-Cre2, all colonies were counted from one plate of each replicate.

**Analysis of yeast plasmids by restriction enzyme digestion**. Plasmids were isolated from yeast colonies using the Zymoprep Yeast Plasmid Miniprep II kit (Zymo Research) and subsequently retransformed into E. coli strain DH5α. After plasmid purification from E. coli, plasmids were digested with PstI and SacI and obtained fragments were separated via agarose gel electrophoresis.

**Flow cytometry**. One ml SC drop-out medium, containing 25 μM PCB in the case of Light-Cre3, was inoculated with 10 μl of overnight culture in 12-well cell culture plates and incubated with shaking for 24 h (30 °C, 200 rpm). After 6 h of growth in the dark, cells were induced with a 5-min red light pulse (Light-Cre3) or 1 μM β-testradiol (EST-Cre3) and maintained for further 24 h in the dark (EST-Cre3) or with 10-s red light pulses every 5 min (Light-Cre3). Afterwards, fluorescence was determined via flow cytometry. Fluorescence output of single cells was determined with a BD FACSCalibur flow cytometer (BD Biosciences, Heidelberg, Germany) with a 488-nm excitation laser and 530/30 nm band pass filter for detection. The cells were diluted in water and passed through the cytometer and 20,000 cells were counted per measurement. To determine the percentage of yEGFP-positive cells, the yEGFP-negative gate was defined to contain at least 99% of BY4742 cells containing pLH_Scr18 (negative control). Cells with yEGFP fluorescence above the threshold were counted as yEGFP-positive cells. All results shown are mean values calculated from three biologically independent colonies, each with three technical replicates. Standard deviation (SD) is given by error bars.

**Light sources**. COMPLED Solutions GmbH (Dresden, Germany) installed LED light sources for red light (λ = 640–680 nm, peak at 661 nm) and far-red light (λ = 720–760 nm, peak at 740 nm) into an INFORS HT Multitron Pro incubator (INFORS AG, Bottmingen, Switzerland). 40 LEDs per wavelength were installed on the ceiling of the incubator; light intensity was 2.8 mW cm$^{-2}$ for red light and 6.9 mW cm$^{-2}$ for far-red light. For uniform illumination, the incubator walls were covered with light-reflecting foil. Light-sensitive experiments in PCB-containing medium were done under green safelight using a standard LED strip.

**Data availability**. Plasmids pLH_Scr15, pLH_Scr16 and pLM494 are available from Addgene (www.addgene.org). All other strains and plasmids reported in this study are available upon request from our laboratory.

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

## Acknowledgements

This research was funded by the Federal Ministry of Education and Research of Germany (BMBF; FKZ 031A172). L.H. thanks the Potsdam Graduate School for financial support and Dr. Fabian Machens for helpful discussions.

## Author contributions

L.H. designed and developed the overall L-SCRaMbLE strategy with contributions from L.A.M. L.H. conducted the experiments and analyzed the data. L.H. wrote the manuscript, with contributions from B.M.-R. and L.A.M. L.A.M. designed and constructed plasmids pLM494, pLM006 and yeast strain yLM1295. K.S. constructed plasmids pLH_Scr12 to pLH_Scr16. B.M.-R. supervised the work. Experiments were mostly done in the Cell2Fab lab, led by K.M. All authors take full responsibility for the content of the paper.

## Additional information

**Competing interests:** The authors declare no competing financial interests.

