## [Peer Review File · Nature Communications]

Reviewers' comments:

Reviewer #1 (Remarks to the Author):

The work of Mueller-Roeber and coworkers demonstrated a clever approach to fine-tune the activity of a split recombinase system for scrambling synthetic yeast chromosomes. In this technology, the Mueller-Roeber group fused split Cre recombinase with a light-inducible dimerization system PhyBNT-PIF3, and activate Cre-mediated recombination with intermittent pulses of red light. Since addition of phycocyanobilin (PCB), chromophore of the PhyB receptor, is required for light-inducible dimerization, PCB concentration can be employed to introduce an additional layer of control over Cre recombinase activity. This is a simple technique to fine-tune Cre activity. However, this reviewer feels that additional work needs to be carried out to further demonstrate that this inducible method is superior to the system of estradiol-inducible Cre-EBD which is currently used in scrambling chromosomes in recent landmark papers. Specifically, there are a few major points the authors need to address:

1. The authors claim that the use of a light-inducible system is needed to overcome the toxicity issue in using estradiol to induce Cre-EBD. However, this claim is opposite to what has been shown by Dymond et al. *Nature* 2011 and many other published works in yeast (e.g. Carlile and Amon *Cell* 2008). Furthermore, the issue with the basal activity in Cre-EBD has been recognized before, and it has been demonstrated that an additional EBD, i.e. EBD-Cre-EBD, also helps minimize the background activity (Matsuda and Cepko *PNAS* 2007). Perhaps the authors need to provide more data on why the light-inducible system provides an advantage in yeast genome engineering projects, especially in the context of large-sized chromosome arms instead of a model 4-gene system.

2. The choice of the light-inducible system. The authors need to substantiate on the reason in adopting the PhyBNT-PIF3 system in their work, rather than other light-inducible dimerization systems such as pMag/nMag (Kawano *Nat. Chem. Biol* 2016) or Cry-CIB (Kennedy *Nat. Meth.* 2010). While considering the requirement of PCB can be considered to be useful to regulate the basal activity of split Cre, the authors need to show a comparison study among the 3 systems. Otherwise, one can consider that this is another light-inducible Cre system in addition to many already existing light- and chemically-inducible systems.

Some minor points:

3. *Nature Communication* publications are in Article formats with headed sections Results, Discussion and Methods.

4. The Methods section in the main text needs further editing, and the reference to *Nature Biotechnology* needs to be replaced.

5. Analysis of yellow vs. white colonies. The author should provide information on the experimental step required to classify the color of these colonies.

6. Supplementary Figure 4. Percentages in English should be written as 0.4% instead of 0,4%.

Reviewer #2 (Remarks to the Author):

Hochrein et al. present L-SCRaMble, a variation of the original SCRaMble tool for Cre-recombination in yeast. The original tool relies on an estradiol-dependent induction of Cre recombinase activity. The authors have modified the induction system in order to gain tighter control of Cre recombinase activity. In this way, they have designed a split-Cre recombinase that is reconstituted upon exposure to red light in the presence of a specific chromophore (PCB) and is

thus activated. Moreover, they have increased the control of Cre recombinase activity by including a nuclear localization signal (NLS) only in one of the two fragments of the split-enzyme. Here, they compare the recombination capacities of their light-induced system to the original estradiol-dependent one by testing both in yeast. Their test system uses a low-copy number plasmid carrying the genes of a β -carotene pathway, each one flanked by loxP sites.

The light-based induction used in L-SCRaMbLE is a great and interesting idea, which I was very keen to read and learn about. It appears to be the first light-induced Cre-recombinase system designed for yeast, although there is (and they do mention it in their text) a similar system available for mammalian cells (ref. 20). The downside, however, is the technical quality of the results and the discussion, which does not correspond with the expectations created when reading the design of their system.

Issues to address before further consideration of this manuscript for publishing:

1. The manuscript needs to be restructured to meet the standards of Nat. Coms. and the text should be rewritten to make it more clear to the reader.

Separate sections including summary, introduction and results/discussion (with subsections) should be addressed. The introduction is not currently accessible, except to specialised readers.

2. In the second paragraph, while describing the split-Cre recombinase system designed, they mention the recently published blue-light responsive split-Cre system for mammalian cells (ref. 20 in the manuscript). The authors of this work claim that >20% recombination rates, and low background, were observed after a single pulse of 4s of blue light. Hochrein et al., have based their current design on this previous system for mammalian cells. They should test the older system in yeast as a benchmark. It seems to be more powerful than the red-light system that they developed, as they need continuous short red-light pulses over several hours to reach the maximum 50% recombination rates that they report here.

3. If the goal is to test the total recombination potential of these systems, why use this complex system based on plasmid pLM494? As they explain on page 3, the recombination possibilities between all loxP sites will result in very many different phenotypes. Nevertheless, they choose a phenotype read out to quantify recombination rates, even stating that there are two cases where it will be underestimated.

In the first part of the work presented, investigating the power of the systems, it would be useful to use a simplified system with plasmids carrying only 2 loxP sites and analyse recombination rates by qPCR.

4. Following this last point; why, in the analysis of recombination rates using the original SCRaMbLE system, do they count so few colonies compared to the other samples? Is the estradiol causing toxicity to the culture? Why did they increase the amount of estradiol needed for induction, from 1 μ M reported to 2 μ M? It would be important to standardise the colony number sampling.

5. Why did they generate strains with different auxotrophies when wanting to compare L-SCRaMbLE to SCRaMbLE?

6. They mention in the manuscript that the recombination rates, especially the basal levels observed, do not match the reported results. However, they do not attempt to address or explain this in the manuscript.

7. At the end of the manuscript, they describe the effect of PCB concentrations on L-SCRaMbLE activity. In this way, they show the dependency of L-SCRaMbLE on PCB (>5 μ M) for activity. Have they tested concentrations higher than 25 μ M PCB? Do they think that this could break the ceiling of 50% recombination rate?

8. In their last paragraph, they state "L-SCRaMbLE allows for tight control of recombination pre-induction and high Cre-mediated recombination after light induction requiring only brief light pulses". I agree, although they should qualify it, adding that short pulses are needed for several hours, which translates into a need for long induction timings (it is also true that the alternative estradiol-system uses also long induction periods).

9. Overall, I agree that L-SCRaMbLE is potentially a great tool, especially as its basal levels are incredibly low and the additional control by PCB makes it even tighter. I only wonder if the

approximate maximum 50% recombination rate makes it powerful enough when applied to the *Saccharomyces cerevisiae* genome re-engineering project Sc2.0.

Moreover, there are some technical issues that need to be addressed as well:

10. When mentioning the "two proteins tested", it would be clarifying to include the names of those proteins.

11. The Suppl. Figs. showing colony plates are very low quality. It is difficult to distinguish between yellow and white colonies (specially in suppl. fig. 1). Also, the number of colonies (and colony sizes) shown is very disparate between samples, making it difficult to make comparisons. Size bars are needed and images showing similar numbers (and sizes) of colonies would be better.

12. Suppl. Fig. 2 and 4: tables should be improved, and include mean values and SD for the bio-replicates presented (which are mentioned in the main text).

13. I would suggest that suppl. Fig. 5 should be shown in the main text.

14. Suppl. Fig. 7 and 8: All maps of plasmids need to be revised to correct names on the labels included (e.g. loxPsym is sometimes called loxP). Also, relevant RE sites and primer binding sites would be good to be indicated on the vector maps.

Figure legends need to include corresponding names of the abbreviations on labels included in the maps of plasmids.

Note: Changes in the manuscript text are shown using the following color code:

BLUE – new text

GREEN – text that was already in the previous version of the manuscript but has been moved to new location now

Reviewers' comments:

Reviewer #1 (Remarks to the Author):

The work of Mueller-Roeber and coworkers demonstrated a clever approach to fine-tune the activity of a split recombinase system for scrambling synthetic yeast chromosomes. In this technology, the Mueller-Roeber group fused split Cre recombinase with a light-inducible dimerization system PhyBNT-PIF3, and activate Cre-mediated recombination with intermittent pulses of red light. Since addition of phycocyanobilin (PCB), chromophore of the PhyB receptor, is required for light-inducible dimerization, PCB concentration can be employed to introduce an additional layer of control over Cre recombinase activity. This is a simple technique to fine-tune Cre activity. However, this reviewer feels that additional work needs to be carried out to further demonstrate that this inducible method is superior to the system of estradiol-inducible Cre-EBD which is currently used in scrambling chromosomes in recent landmark papers. Specifically, there are a few major points the authors need to address:

1. The authors claim that the use of a light-inducible system is needed to overcome the toxicity issue in using estradiol to induce Cre-EBD. However, this claim is opposite to what has been shown by Dymond et al. Nature 2011 and many other published works in yeast (e.g. Carlile and Amon Cell 2008). Furthermore, the issue with the basal activity in Cre-EBD has been recognized before, and it has been demonstrated that an additional EBD, i.e. EBD-Cre-EBD, also helps minimize the background activity (Matsuda and Cepko PNAS 2007). Perhaps the authors need to provide more data on why the light-inducible system provides an advantage in yeast genome engineering projects, especially in the context of large-sized chromosome arms instead of a model 4-gene system.

RESPONSE:

We think there is a misunderstanding; we meant the toxicity of beta-estradiol for the experimenter, not the yeast cells. In that respect, a light-inducible system would be more user friendly as it is easier to handle.

However, this was not the main reason for establishing a light-mediated recombination system in yeast. Importantly, we wanted to build a system that allows fine-tuning of the recombination efficiency simply by modifying the induction time and/or the PCB concentration, and the reliable switch to an off-state by withdrawing PCB. Even if lower basal expression can also be achieved by engineering the estradiol-inducible system, reversible induction is difficult to achieve with a chemical inducer. With light induction, the system can be switched on and off simply by changing the wavelength of the light source.

Furthermore, we could show that L-SCRaMBLE achieves a greater diversity of recombination events. In our opinion, a system mediating smooth recombination is advantageous over the estradiol-inducible system which shows 100% plasmids with complete deletion of all 4 genes and in this way no variety at all (Figure 3). When scrambling the synthetic chromosomes harsh deletions would cause

cell death. A variety of small deletions would be beneficial to create an optimized host organism for the improved production of heterologous metabolites or for the establishment of a minimal genome.

We made various changes on numerous parts of the manuscript to address the reviewer's comment.

2. The choice of the light-inducible system. The authors need to substantiate on the reason in adopting the PhyBNT-PIF3 system in their work, rather than other light-inducible dimerization systems such as pMag/nMag (Kawano Nat. Chem. Biol 2016) or Cry-CIB (Kennedy Nat. Meth. 2010). While consider the requirement of PCB can be considered to be useful to regulate the basal activity of split Cre, the authors need to show a comparison study among the 3 systems. Otherwise, one can consider that this is another light-inducible Cre system in addition to many already existing light- and chemically-inducible systems.

RESPONSE:

The reviewer is right, there are different blue light-inducible systems established in mammalian cells that show good results. However, to our knowledge, L-SCRaMbLE represents the first light-induced Cre-recombinase for use in *S. cerevisiae*.

As the reviewer pointed out, the established blue light-inducible systems for mammalian cells incorporate chromophores available in yeast. In this way, fine-tuning by changing chromophore concentration or a complete shut-down by withdrawal of the chromophore is not possible. Furthermore, blue light shows toxic effects on yeast cells and restricts the dose of applied blue light (Robertson et al., Proceedings of the National Academy of Sciences USA, 2013). In contrast, red light was shown to have only negligible effects on yeast metabolism when applied in high intensity (Robertson et al., Proceedings of the National Academy of Sciences USA, 2013).

A direct comparison of the three systems in yeast cannot be carried out in a reasonable time scale, as the different blue light systems established for recombination in mammalian cells need to be carefully adopted for expression in yeast. As the expression strengths of the optical dimerizer is a critical step for the establishment of any light-induction system, a lengthy optimization process would be needed. However, to put our results in a similar context as the systems described in Kawano Nat. Chem. Biol 2016 and Kennedy Nat. Meth. 2010 to L-SCRaMbLE, we designed plasmid pLH_Scr18, containing a *loxP*-flanked terminator cassette upstream of the yEGFP reporter. When comparing the recombination efficiencies achieved in yeast cells by L-SCRaMbLE with the blue light systems in mammalian cells it becomes obvious that L-SCRaMbLE shows a comparable outcome. The split Cre system established by Kennedy (Kennedy *et al.*, Nat. Meth. 2010) achieved recombination efficiencies of 16.4% when mammalian cells were exposed to continuous blue light pulses for 24 hours. L-SCRaMbLE generates around 12% recombined cells after 24 h incubation in inducing conditions and 25 μ M PCB in the medium (Fig. 5). Furthermore, we now demonstrate in an additional experiment, that increasing the PCB concentration in the medium to up to 100 μ M PCB can raise recombination efficiency to up to 90% (compared to 47% with 25 μ M PCB) after 16 h of growth in inducing conditions, when applied to plasmid pLM494 containing four *loxP**sym*-flanked genes. While Kawano (Kawano et al., Nat. Chem. Biol., 2016) reported 320-fold increase of recombination upon induction with the Magnet system, L-SCRaMbLE achieves 180-fold induction of recombination over background level, when tested with plasmid pLM494 (89,5% recombination with 100 μ M PCB after 16 h induction vs. 0.5% recombination efficiencies without PCB; see Suppl. Fig. 4).

Compared to the established estradiol-inducible SCRaMbLE system, L-SCRaMbLE reported here achieves a wide range of different recombination outcomes (Fig. 3) and recombination efficiency depends on the red light induction time and the PCB concentration used (Fig. 2 and 4). This is a

big advantage for the synthetic yeast project, as harsh deletions in the genome would cause cell death.

We added the following section to the manuscript (page 13):

“While L-SCRaMbLE is the first light-regulated Cre recombinase for use in yeast, several blue light-responsive systems are available for mammalian cells that are based on plant-derived CRY2-CIB1 proteins^{18,19} or the Magnet system, engineered from a fungal photoreceptor³⁷. Regarding the fold-induction or the percentage of recombination activity reported for those systems, L-SCRaMbLE shows comparable, or even better results depending on the reporter system. However, while single blue light pulses are sufficient to activate Cre recombinase in mammalian systems, L-SCRaMbLE depends on repetitive red light pulses applied over a longer period. Although this may appear unfavorable at first glance, a red light-controlled recombination system bears considerable advantages in yeast. First, the current blue light-inducible systems depend on a chromophore naturally present in yeast, precluding fine-tuning of recombinase activity by varying chromophore concentration. Second, a complete shut-off, independent of light status, is not possible. Third, blue light affects yeast growth and metabolism, while red light only negligibly affects yeast physiology³⁸. This eliminates the need for a system that fully responds to a single red light pulse. Fourth, multi-pulse induction over several hours is highly beneficial for a light-inducible SCRaMbLE system. While short pulses are advantageous for applications in mammalian cells, induction with a single pulse in yeast would result in few recombination events that are genetically inherited over the time span of the induction experiment. When a single pulse is applied to the samples, PhyB will dimerize with PIF3 and the functionally reconstituted Cre recombinase can act on *loxPsym* sites. However, after some time in darkness, PhyB will return to its Pr state and the PhyB-PIF3 dimer will dissociate, leaving Cre recombinase non-functional. Thereafter, recombination events will be inherited approximately every 1.5 h due to cell division, without creating new recombinations. Naturally, photoreceptor proteins expressed after single pulse induction will also be inactive. As generation times of mammalian cells are much longer compared to yeast, this effect might be negligible in those systems. Furthermore, while the tools established for mammalian cells intended to generate single specific recombination outcomes, SCRaMbLE-ing a synthetic yeast genome aims at generating numerous independent and diverse recombination events. This can only be achieved with moderate recombination activity acting over various periods of time, as made possible with L-SCRaMbLE reported here.”

Some minor points:

3. Nature Communication publications are in Article formats with headed sections Results, Discussion and Methods.

RESPONSE:

We changed the manuscript accordingly.

4. The Methods section in the main text needs further editing, and the reference to Nature Biotechnology needs to be replaced.

RESPONSE:

The reference to Nature Biotechnology was replaced and the Methods section was extended.

5. Analysis of yellow vs. white colonies. The author should provide information on the experimental step required to classify the color of these colonies.

RESPONSE:

We added the following text to the method section of the manuscript (page 17):

“Analysis of recombination experiments

For experiments carried out with Light-Cre1, Light-Cre2 and EST-Cre1 cells, recombination efficiency was calculated by the ratio of white to yellow colonies. To this end, white and yellow colonies of one plate of each replicate were counted. Orange or slightly yellow colonies were counted as yellow colonies. Mixed colonies showing white and yellow sectors were counted as white, as well as yellow colonies. For experiments carried out with yeast strain EST-Cre2, all colonies were counted from one plate of each replicate.”

6. Supplementary Figure 4. Percentages in English should be written as 0.4% instead of 0,4%.

RESPONSE:

Thanks for pointing this out. We corrected it.

Reviewer #2 (Remarks to the Author):

Hochrein et al. present L-SCRaMbLE, a variation of the original SCRaMbLE tool for Cre-recombination in yeast. The original tool relies on an estradiol-dependent induction of Cre recombinase activity. The authors have modified the induction system in order to gain tighter control of Cre recombinase activity. In this way, they have designed a split-Cre recombinase that is reconstituted upon exposure to red light in the presence of a specific chromophore (PCB) and is thus activated. Moreover, they have increased the control of Cre recombinase activity by including a nuclear localization signal (NLS) only in one of the two fragments of the split-enzyme. Here, they compare the recombination capacities of their light-induced system to the original estradiol-dependent one by testing both in yeast. Their test system uses a low-copy number plasmid carrying the genes of a β -carotene pathway, each one flanked by loxPsym sites.

The light-based induction used in L-SCRaMbLE is a great and interesting idea, which I was very keen to read and learn about. It appears to be the first light-induced Cre-recombinase system designed for yeast, although there is (and they do mention it in their text) a similar system available for mammalian cells (ref. 20). The downside, however, is the technical quality of the results and the discussion, which does not correspond with the expectations created when reading the design of their system.

Issues to address before further consideration of this manuscript for publishing:

1. The manuscript needs to be restructured to meet the standards of Nat. Coms. and the text should be rewritten to make it more clear to the reader.

Separate sections including summary, introduction and results/discussion (with subsections) should be addressed. The introduction is not currently accessible, except to specialised readers.

RESPONSE:

We restructured the text to meet the Nature Communications standard. We rewrote the Introduction to make it more accessible to a wider audience. The text was rewritten to make it clearer to the reader.

2. In the second paragraph, while describing the split-Cre recombinase system designed, they mention the recently published blue-light responsive split-Cre system for mammalian cells (ref. 20 in the manuscript). The authors of this work claim that >20% recombination rates, and low background, were observed after a single pulse of 4s of blue light. Hochrein et al., have based their current design on this previous system for mammalian cells. They should test the older system in yeast as a benchmark. It seems to be more powerful than the red-light system that they developed, as they need continuous short red-light pulses over several hours to reach the maximum 50% recombination rates that they report here.

RESPONSE:

In our opinion induction with a single pulse would not be beneficial in our experiments. The reviewer is right, Taslimi et al., 2016 (ref. 20), showed around 20% recombination rate after induction with a single 4-sec blue light pulse and incubation for 24 h. While short single pulses are advantageous for applications in mammalian cells, induction with a single red light pulse in yeast would result in few recombination events that are inherited over the incubation time. When a single pulse is applied to the samples, PhyB will dimerize with PIF3 and the functional Cre recombinase can act on loxP sites. However, after a certain time in darkness PhyB will return to its Pr state and the dimer will dissociate. In this way, the Cre recombinase is not functional anymore. From that moment on the recombination events will be inherited approximately every 1.5 h due to cell division without creating new recombination events. As generation times of mammalian cells are much longer compared to yeast, this effect might be negligible in those systems. Furthermore, while the systems established in mammalian cells aimed to generate a single specific recombination event, scrambling a synthetic yeast genome requires numerous independent and different recombination events. This can only be achieved with moderate recombination efficiency over various times. Therefore, with L-SCRaMbLE, we did not focus on single pulse efficiencies, but on diverse recombination outcome, tunability of recombinase activity and reliable off-state.

In this context, it should be mentioned that the choice of red light is an important aspect as red light was shown to have negligible effect on yeast cells, while blue light negatively affects yeast cells (Robertson et al., Proceedings of the National Academy of Sciences USA, 2013).

In order to compare recombination efficiency of L-SCRaMbLE with the efficiency achieved by the CRY2-CIB1 based Cre recombinase, reported by Taslimi *et al*, we designed plasmid pLH_Scr18 which contains a loxP-flanked terminator cassette upstream of the reporter yEGFP. Regarding recombination efficiencies, L-SCRaMbLE generates a recombination efficiency similar to those reported by Taslimi *et al*. after 24 h incubation in inducing conditions and 25 μ M PCB in the media (Fig. 5). A direct comparison of L-SCRaMbLE with the earlier system published by Taslimi *et al*. cannot be carried out in a reasonable time scale, as the promoters driving the expression of the blue light sensing optical dimerizer need to be exchanged. As the expression strengths of the optical dimerizer is a critical step for the establishment of any light induction system, a lengthy optimization process would be needed.

Furthermore, in an additional experiment we now demonstrated, that increasing the PCB concentration in the medium to up to 100 μ M can raise recombination efficiency from around 47% with 25 μ M PCB, up to 90%, when applied to plasmid pLM494, containing four *loxPsym*-flanked genes. However, a high recombination efficiency like in the EST-Cre system is not desirable for SCRaMbLE experiments, as such high activity causes harsh deletions resulting in uniform plasmids with simultaneous deletions of all four *loxPsym*-flanked genes (Fig. 3). Harsh deletions are expected to cause death of cells carrying synthetic genomes. A variety of small deletions, however, would be beneficial to create an optimized host organism for the improved production of heterologous metabolites or for the establishment of a minimal genome.

We added the following section to the manuscript (page 13):

“While L-SCRaMbLE is the first light-regulated Cre recombinase for use in yeast, several blue light-responsive systems are available for mammalian cells that are based on plant-derived CRY2-CIB1 proteins^{18,19} or the Magnet system, engineered from a fungal photoreceptor³⁷. Regarding the fold-induction or the percentage of recombination activity reported for those systems, L-SCRaMbLE shows comparable, or even better results depending on the reporter system. However, while single blue light pulses are sufficient to activate Cre recombinase in mammalian systems, L-SCRaMbLE depends on repetitive red light pulses applied over a longer period. Although this may appear unfavorable at first glance, a red light-controlled recombination system bears considerable advantages in yeast. First, the current blue light-inducible systems depend on a chromophore naturally present in yeast, precluding fine-tuning of recombinase activity by varying chromophore concentration. Second, a complete shut-off, independent of light status, is not possible. Third, blue light affects yeast growth and metabolism, while red light only negligibly affects yeast physiology³⁸. This eliminates the need for a system that fully responds to a single red light pulse. Fourth, multi-pulse induction over several hours is highly beneficial for a light-inducible SCRaMbLE system. While short pulses are advantageous for applications in mammalian cells, induction with a single pulse in yeast would result in few recombination events that are genetically inherited over the time span of the induction experiment. When a single pulse is applied to the samples, PhyB will dimerize with PIF3 and the functionally reconstituted Cre recombinase can act on *loxPsym* sites. However, after some time in darkness, PhyB will return to its Pr state and the PhyB-PIF3 dimer will dissociate, leaving Cre recombinase non-functional. Thereafter, recombination events will be inherited approximately every 1.5 h due to cell division, without creating new recombinations. Naturally, photoreceptor proteins expressed after single pulse induction will also be inactive. As generation times of mammalian cells are much longer compared to yeast, this effect might be negligible in those systems. Furthermore, while the tools established for mammalian cells intended to generate single specific recombination outcomes, SCRaMbLE-ing a synthetic yeast genome aims at generating numerous independent and diverse recombination events. This can only be achieved with moderate recombination activity acting over various periods of time, as made possible with L-SCRaMbLE reported here.”

3. If the goal is to test the total recombination potential of these systems, why use this complex system based on plasmid pLM494? As they explain on page 3, the recombination possibilities between all *loxPsym* sites will result in very many different phenotypes. Nevertheless, they choose a phenotype read out to quantify recombination rates, even stating that there are two cases where it will be underestimated.

In the first part of the work presented, investigating the power of the systems, it would be useful to

use a simplified system with plasmids carrying only 2 loxPsym sites and analyse recombination rates by qPCR.

RESPONSE:

We agree that counting the ratio of white to yellow colonies mediated upon recombination of pLM494 is not very reliable. We used this experimental setup to investigate the variety of possible outcome and the tunability of L-SCRaMbLE, which are very important criteria for scrambling the synthetic yeast genomes. We now added a further experiment with an experimental setup comparable to the system used by Taslimi et al. We established plasmid pLH_Scr18 which contains a *loxP*-flanked terminator upstream of the γ EGFP reporter. Using this setup, we observed recombination efficiencies of around 12% after 24 h induction and 0.6% in uninduced conditions.

We added Results section “**Specific recombination efficiency towards a *loxP*-flanked DNA fragment**” to pages 13 and 14 and extended the Discussion accordingly.

4. Following this last point; why, in the analysis of recombination rates using the original SCRaMbLE system, do they count so few colonies compared to the other samples? Is the estradiol causing toxicity to the culture? Why did they increase the amount of estradiol needed for induction, from 1 μ M reported to 2 μ M? It would be important to standardise the colony number sampling.

RESPONSE:

We always plated the same dilution of each sample (Light-Cre1,2 and EST-Cre) and counted one plate. As we obtained less colonies for EST-Cre, we also counted less colonies for EST-Cre. However, we now adjusted the number of counted colonies of EST-Cre, as suggested by the reviewer. As the colony numbers were comparable for induced and uninduced samples, we do not think that estradiol is causing toxicity (at the concentration used). We also had a look at the OD measured before plating the cells and these values are slightly lower compared to Light-Cre samples (see figure). Nevertheless, EST-Cre samples do just serve as a control in our experiments. We think that the difference in colony number is no serious issue and hope the reviewer agrees with it.

The reviewer is right, we induced the samples with 2 μM β -estradiol compared to the original SCRaMbLE protocols which report induction with 1 μM β -estradiol. We therefore repeated the experiments with 1 μM estradiol and observed that the difference between the data for achieved recombination efficiencies are negligible. However, we exchanged the data in the manuscript. Thanks to the reviewer for pointing this out.

5. Why did they generate strains with different auxotrophies when wanting to compare L-SCRaMbLE to SCRaMbLE?

RESPONSE:

We established the L-SCRaMbLE system with the auxotrophic marker Leu as L-SCRaMbLE is based on our previously reported PhiReX system for red light-induced gene expression, that also works with a Leu marker (ref. 21 of our manuscript; Hochrein *et al.*, 2017, *Nucleic Acids Res* **45**, 9193-9205, doi:10.1093/nar/gkx610). As we aimed at comparing L-SCRaMbLE and SCRaMbLE with regard to the achieved recombination efficiency and possibilities for fine-tuning and regulation rather than with respect to the molecular design, we do not see a problem in the selection of a different auxotroph marker compared to the original SCRaMbLE system. In particular, the synthetic Sc2.0 yeast strains do not contain the LEU marker ensuring compatibility with L-SCRaMbLE. We hope the reviewer agrees with our opinion in this point.

6. They mention in the manuscript that the recombination rates, especially the basal levels observed, do not match the reported results. However, they do not attempt to address or explain this in the manuscript.

RESPONSE:

To assess the background activity of the estradiol-inducible Cre recombinase in a genome-integrated context, we now integrated a *loxP*-flanked *URA3* cassette into the *HO* locus of the yeast genome. *LoxP* sites are oriented in a way that Cre recombinase activity should cause deletion of the *URA3* cassette. We transformed the newly generated strain with pLM006 and plated induced and uninduced samples on SD media with and without 5-FOA for counter selection of *URA3* expression (while 5-FOA is nontoxic to yeast in the absence of *URA3*, 5-FOA is converted to the toxic form when the *URA3* gene is expressed).

For β -estradiol-induced samples we obtained recombination efficiencies of 6.5% after 24 h of induction. For uninduced samples we observed background recombination of 1.4%. This clearly demonstrates, that Cre recombinase acts less efficiently on genome-integrated *loxP* sites compared to episomal ones (94% recombination efficiency after induction and 40% recombination without induction, see Fig. 2). Therefore, it is explainable why Shen *et al.* (Shen *et al.*, Genome research, 2016) reported no background recombination with the synthetic yeast chromosome *synIXR*. Remarkably, *synIXR* has only 43 *loxP*sym sites, while the final synthetic yeast genome will incorporate 4000 *loxP*sym sites. This makes background recombination much more likely in the final (complete) yeast synthetic genome and highlights the advantage of L-SCRaMbLE.

We added the last section of the response to the Discussion (page 12) and the following section to the Results (page 5):

“As a high basal activity of the EST-Cre system has not been reported within the context of a circular Sc2.0 chromosome arm, *synIXR*³⁴, which is ~90 kb long and encodes 43 *loxP*sym sites, we integrated two *loxP* sites, flanking an *URA3* cassette, into the *HO* locus of the yeast genome, resulting in strain

yLM1295. After transformation with plasmid pLM006, the newly generated strain EST-Cre2 was tested with regard to the recombination activity between two *loxP* sites after β -estradiol induction. Successful recombination will cause deletion of the *URA3* cassette. Recombination activity was evaluated by plating samples on SD media with or without 5-fluoroorotic acid (FOA) for counter selection of *URA3* expression (**Supplementary Fig. 2A**). While cells with a successfully deleted *URA3* gene grow on FOA-containing medium, cells expressing the *URA3* gene do not. Estradiol-induced samples showed recombination efficiencies of only 6.5% after 24 h of induction, while uninduced samples showed 1.4% background recombination (**Supplementary Fig. 2B and C**). This demonstrates that Cre recombinase acts less efficiently on two genome-integrated *loxP* sites than on multiple episomal *loxPsym* sites (100% recombination activity after already 16 h incubation for induced samples, and 45% for uninduced ones, **Fig. 2**). As recombination events between two *loxP* or two *loxPsym* sites were shown to occur with approximately equal efficiency, at least in *Escherichia coli* cells, we assume that the observed difference in recombination efficiency is not due to the spacer region within the recombination sites¹⁵. As the observed recombination efficiencies of the light-inducible systems Light-Cre1 and Light-Cre2 were up to 20 times lower than in the β -estradiol inducible system EST-Cre1, when acting on pLM494, the number of FOA resistant clones generated in the context of two genome-integrated *loxP* sites was too low for a reliable evaluation of recombination efficiency.”

7. At the end of the manuscript, they describe the effect of PCB concentrations on L-SCRaMbLE activity. In this way, they show the dependency of L-SCRaMbLE on PCB (>5 μ M) for activity. Have they tested concentrations higher than 25 μ M PCB? Do they think that this could break the ceiling of 50% recombination rate?

RESPONSE:

Considering the reviewer’s suggestion, we tested L-SCRaMbLE with higher PCB concentrations of 50 μ M and 100 μ M, respectively. We could show that this boosts recombination efficiencies to 64 % (50 μ M PCB) and even 90 % (100 μ M PCB). However, also basal levels increase with higher PCB concentrations. In our opinion, this basal activity can be neglected as SCRaMbLE experiments are typically only conducted in PCB-containing medium, when recombination is intended. If not, the user would maintain the synthetic strain in medium without PCB and observe no background activity. We added the new data to Fig. 4 and changed the text accordingly (page 11).

8. In their last paragraph, they state “L-SCRaMbLE allows for tight control of recombination pre-induction and high Cre-mediated recombination after light induction requiring only brief light pulses”. I agree, although they should qualify it, adding that short pulses are needed for several hours, which translates into a need for long induction timings (it is also true that the alternative estradiol-system uses also long induction periods).

RESPONSE:

We agree with the reviewer’s opinion and deleted the part “only brief light pulses”.

9. Overall, I agree that L-SCRaMbLE is potentially a great tool, especially as its basal levels are incredibly low and the additional control by PCB makes it even tighter. I only wonder if the approximate maximum 50% recombination rate makes it powerful enough when applied to the *Saccharomyces cerevisiae* genome re-engineering project Sc2.0.

RESPONSE:

We showed that L-SCRaMbLE allows precise fine-tuning of recombination events and attaining a reliable off-state. We think, that this is more important than a high recombination rate (even if we could show recombination efficiencies of 90% with 100 μ M PCB), as harsh deletions of the synthetic genome would cause cell death. Further, small and diverse changes mediated by systems generating mediate recombination efficiencies, like L-SCRaMbLE, increase the opportunity for creating optimized yeast strains. We considerably expanded the discussion to address this point.

Moreover, there are some technical issues that need to be addressed as well:

10. When mentioning the “two proteins tested”, it would be clarifying to include the names of those proteins.

RESPONSE:

We changed the text accordingly (page 1).

11. The Suppl. Figs. showing colony plates are very low quality. It is difficult to distinguish between yellow and white colonies (specially in suppl. fig. 1). Also, the number of colonies (and colony sizes) shown is very disparate between samples, making it difficult to make comparisons. Size bars are needed and images showing similar numbers (and sizes) of colonies would be better.

RESPONSE:

We changed the figures according to the reviewer’s suggestions.

12. Suppl. Fig. 2 and 4: tables should be improved, and include mean values and SD for the bio-replicates presented (which are mentioned in the main text).

RESPONSE:

Suppl. Fig. 1 (which the reviewer probably meant) and Suppl. Fig. 6 (former Suppl. Fig 4) were improved according to the reviewer’s suggestions.

13. I would suggest that suppl. Fig. 5 should be shown in the main text.

RESPONSE:

We included Suppl. Fig. 5 in the main text as Figure 4 (page 12).

14. Suppl. Fig. 7 and 8: All maps of plasmids need to be revised to correct names on the labels included (e.g. loxPsym is sometimes called loxP). Also, relevant RE sites and primer binding sites would be good to be indicated on the vector maps. Figure legends need to include corresponding names of the abbreviations on labels included in the maps of plasmids.

RESPONSE:

Vector maps in Suppl. Fig. 7 and 8 were changed according to the reviewer’s suggestions. Names of abbreviations are now included in the figure legends.

REVIEWERS' COMMENTS:

Reviewer #1 (Remarks to the Author):

The work by the Mueller-Roeber group showed a robust method for scrambling yeast chromosomes via a photo-activatable version of split Cre recombinase. Their L-SCRaMble system requires an addition of phycocyanobilin (PCB) for light-inducible dimerization of split Cre halves and is able to exhibit low basal activity of Cre-mediated recombination. With different concentrations of PCB and duration of light induction, Hochrein et al. were able to fine-tune the activity of split Cre recombinase, demonstrating the utility of L-SCRaMble as a tool to achieve diverse outcomes of genome evolution. The manuscript has been significantly improved compared to the previously submitted one and should be published in Nature Communications. There are still a few minor points that need to be addressed.

1. The choice between Light-Cre1 and Light-Cre2. The basal activity of Light-Cre2 appears to be higher than that of Light-Cre1 at 25 μ M PCB. However, Light-Cre1 is also significantly more active upon 4h- or 16h- light induction. As PCB concentrations can be used to fine-tune split Cre activity, this reviewer believes that Light-Cre2 should actually be a better choice to achieve a wider range of Cre activity for achieving diverse recombination events.
2. Figure 1. Please provide labels for cellular compartments, i.e. nucleus and cytoplasm, as well as clarify which system corresponds to Light-Cre1 or Light-Cre2.
3. Line 119. Please provide an additional schematic on the beta-carotene reporter system and the expected outcomes of recombination events. It is difficult to process through the section without referring to Figure 3.
4. Figure 2. Please provide the definition of the y-axis in the figure. In addition, the concentration of PCB needs to be stated.
5. Line 207. The authors claim that eight different recombination events were detected. However, there seems to be a discrepancy in Figure 3B as the plasmid recovered from colony 7 was not confirmed by sequencing.
6. Line 327. Please explain the difference between Light-Cre3 and the two systems described in Figure 1, i.e. Light-Cre1 and Light-Cre2. Did the authors extend the use of Light-Cre1 to the yEGFP assay of loxP-flanked recombination?

Reviewer #2 (Remarks to the Author):

In this second draft, Hochrein et al. have improved the quality of work presented. They have also improved the manuscript by organising the text into sections. Although, in my opinion, the text is incredibly long and it still needs some reorganisation of the sub-section of results to make it more coherent. It is not optimally publishable in this form.

That said, I believe Hochrein et al. have addressed very well all the points made about the first draft submitted and I agree with their comments. Specially, I think they have addressed well the answer for point 2 with their arguments and agree it is not reasonable to ask for the extra experimental work.

Remaining points:

1. Hochrein et al. presented L-SCRaMble, a fine and elegant system for a tight-control and fine-tuning of Cre-recombination in *Saccharomyces cerevisiae*. They have designed the first light

induced Cre-recombinase system for yeast. I do like specially the fact that the system is virtually silenced in the absence of PCB and that it is possible to reach different levels of recombination depending on PCB concentration and light exposure. However, when comparing the recombination efficiency of their system to the original SCRaMbLE (estradiol induced Cre-recombination) they have shown a poor recombination level (12%). Could this also be improved with increasing concentrations of PCB?

2. Recombination percentages reported in the manuscript are a bit confusing (e.g. Fig. 2C induced vs. Fig. 5B light-cre3 - why the difference?). Is it possible that there is different recombination efficiencies depending on loxP and loxPsym in yeast?

3. They claim that they have demonstrated that "Cre recombinase acts less efficiently on two genome-integrated loxP sites than on multiple episomal loxPsym sites" [based on Est-Cre system]. Will this be the same case for light-Cre? In this way, will light-Cre be at all efficient when applied to SCRaMbLE genome Sc2.0?

REVIEWERS' COMMENTS:

Reviewer #1 (Remarks to the Author):

The work by the Mueller-Roeber group showed a robust method for scrambling yeast chromosomes via a photo-activatable version of split Cre recombinase. Their L-SCRaMbLE system requires an addition of phycocyanobilin (PCB) for light-inducible dimerization of split Cre halves and is able to exhibit low basal activity of Cre-mediated recombination. With different concentrations of PCB and duration of light induction, Hochrein et al. were able to fine-tune the activity of split Cre recombinase, demonstrating the utility of L-SCRaMbLE as a tool to achieve diverse outcomes of genome evolution. The manuscript has been significantly improved compared to the previously submitted one and should be published in Nature Communications. There are still a few minor points that need to be addressed.

1. The choice between Light-Cre1 and Light-Cre2. The basal activity of Light-Cre2 appears to be higher than that of Light-Cre1 at 25 μ M PCB. However, Light-Cre1 is also significantly more active upon 4h- or 16h- light induction. As PCB concentrations can be used to fine-tune split Cre activity, this reviewer believes that Light-Cre2 should actually be a better choice to achieve a wider range of Cre activity for achieving diverse recombination events.

RESPONSE: The reviewer is right, PCB concentration can be used to fine-tune Cre activity. In this way, basal recombination mediated by Light-Cre2 can generally be neglected. However, we aimed at establishing a tool that allows controlling Cre recombinase activity not only by PCB concentration, but also in a time-controlled manner. In this regard, Light-Cre1 is in our opinion the better choice, as Light-Cre1 achieves a wider range of recombination efficiency (14% after 4 h induction and 52% after 24 h induction compared to 27% after 4 h and 53% after 24 h achieved by Light-Cre2). Nevertheless, the future user can flexibly choose which of the L-SCRaMbLE systems fits his/her specific experiment the best.

2. Figure 1. Please provide labels for cellular compartments, i.e. nucleus and cytoplasm, as well as clarify which system corresponds to Light-Cre1 or Light-Cre2.

RESPONSE: We included this information in the figure.

3. Line 119. Please provide an additional schematic on the beta-carotene reporter system and the expected outcomes of recombination events. It is difficult to process through the section without referring to Figure 3.

RESPONSE: We made a few changes in the Results section of the manuscript to better explain the recombination process of plasmid pLM494 (page 4, lower part). Furthermore, we added a new figure as Supplementary Figure 1. All other supplementary figures were renumbered accordingly.

4. Figure 2. Please provide the definition of the y-axis in the figure. In addition, the concentration of PCB needs to be stated.

RESPONSE: We added a sentence to the figure legend to inform about the PCB concentration (25 μ M). At the Y axes, we changed the labelling from “%” to “% of cells”.

5. Line 207. The authors claim that eight different recombination events were detected. However, there seems to be a discrepancy in Figure 3B as the plasmid recovered from colony 7 was not confirmed by sequencing.

RESPONSE: As shown in Figure 3B, eight different recombination events were recovered from the Light-Cre1-induced recombination; all eight recombination results are shown in the figure. Plasmid recovered from colony 7 was not sequenced, as plasmid 3 was sequenced and both plasmids show the same restriction pattern.

6. Line 327. Please explain the difference between Light-Cre3 and the two systems described in Figure 1, i.e. Light-Cre1 and Light-Cre3. Did the authors extend the use of Light-Cre1 to the yEGFP assay of loxP-flanked recombination?

RESPONSE: While strain Light-Cre1 contains plasmid pLH_Scr15 and reporter pLM494, strain Light-Cre3 contains plasmid pLH_Scr15 and reporter plasmid pLH_Scr18. The reviewer is right, we tested pLH_Scr15 from Light-Cre1 with a *loxP*-flanked yEGFP. We added a sentence to the Results section of the main text (page 10, lower part).

Reviewer #2 (Remarks to the Author):

In this second draft, Hochrein et al. have improved the quality of work presented. They have also improved the manuscript by organising the text into sections. Although, in my opinion, the text is incredibly long and it still needs some reorganisation of the sub-section of results to make it more coherent. It is not optimally publishable in this form.

RESPONSE: We checked the length of our manuscript and with its ca. 4,800 words it is still below the length accepted by Nature Communications (5,000 words). We also kept the structure of the manuscript to best guide through the experiments we did. We hope, this is acceptable to the reviewer and journal.

That said, I believe Hochrein et al. have addressed very well all the points made about the first draft submitted and I agree with their comments. Specially, I think they have addressed well the answer for point 2 with their arguments and agree it is not reasonable to ask for the extra experimental work.

Remaining points:

1. Hochrein et al. presented L-SCRaMble, a fine and elegant system for a tight-control and fine-tuning of Cre-recombination in *Saccharomyces cerevisiae*. They have designed the first light induced Cre-recombinase system for yeast. I do like specially the fact that the system is virtually silenced in the absence of PCB and that it is possible to reach different levels of recombination depending on PCB concentration and light exposure. However, when comparing the recombination efficiency of their system to the original SCRaMble (estradiol induced Cre-recombination) they have shown a poor recombination level (12%). Could this also be improved with increasing concentrations of PCB?

RESPONSE: We assume that the lower recombination level in Light-Cre3 compared to Light-Cre1 results from the availability of only two recombination sites in Light-Cre3 compared to five recombination sites in the reporter of Light-Cre1. We are confident, that the recombination efficiency of Light-Cre3 can be increased with increasing PCB concentrations, as both systems rely on the same light-regulated Cre recombinase (pLH_Scr15).

2. Recombination percentages reported in the manuscript are a bit confusing (e.g. Fig. 2C induced vs. Fig. 5B light-cre3 - why the difference?). Is it possible that there is different recombination efficiencies depending on loxP and loxPsym in yeast?

RESPONSE: As already described in response 1, the difference in recombination efficiency of Light-Cre1 and Light-Cre3 likely results from the number of recombination sites integrated in the reporter plasmid. In a previous study, it was shown that Cre recombinase acts with similar efficiencies on symmetrical *loxPsym* and wild-type *loxP* sites *in vitro* (Hoess *et al.*, Nucleic Acids Research, 1986). We added this information to the Discussion of the main text (page 12 bottom / page 13 top).

3. They claim that they have demonstrated that “Cre recombinase acts less efficiently on two genome-integrated loxP sites than on multiple episomal loxPsym sites” [based on Est-Cre system]. Will this be the same case for light-Cre? In this way, will light-Cre be at all efficient when applied to SCRaMbLE genome Sc2.0?

RESPONSE: We think that the light-induced Cre recombinase will also act less efficiently on genome-integrated *loxP* sites, as this difference most likely depends on the action mechanism of Cre and not on the induction system. However, as the synthetic Sc2.0 genome will contain around 4000 *loxPsym* sites, a lower recombination efficiency is straightforward, as small but very diverse recombination events will result from this moderate recombination frequency. Importantly, the recombination efficiency of L-SCRaMbLE can be upregulated by increasing PCB concentration.